# MME-CoT: Benchmarking Chain-of-Thought in Large Multimodal Models for Reasoning Quality, Robustness, and Efficiency

Dongzhi Jiang [* 1]  Renrui Zhang [* † 1]  Ziyu Guo [2]  Yanwei Li [‡ 3]  Yu Qi [‡ 4]  Xinyan Chen [‡ 1]  Liuhui Wang [‡ 5]
Jianhan Jin [‡ 6]  Claire Guo [‡ 7]  Shen Yan [8]  Bo Zhang [8]  Chaoyou Fu [6]  Peng Gao [8]  Hongsheng Li [1 8 9]

## Abstract

Answering questions with Chain-of-Thought (CoT) has significantly enhanced the reasoning capabilities of Large Language Models (LLMs), yet its impact on Large Multimodal Models (LMMs) still lacks a systematic assessment and in-depth investigation. In this paper, we introduce **MME-CoT**, a specialized benchmark evaluating the CoT reasoning performance of LMMs, spanning six domains: math, science, OCR, logic, space-time, and general scenes. As the first comprehensive study in this area, we propose a thorough evaluation suite incorporating three novel metrics that assess the reasoning quality, robustness, and efficiency at a fine-grained level. Leveraging curated high-quality data and a unique evaluation strategy, we conduct an in-depth analysis of state-of-the-art LMMs, uncovering several key insights: *1)* Models with reflection mechanism demonstrate a superior CoT quality, with Kimi k1.5 outperforming GPT-4o and demonstrating the highest quality results; *2)* CoT prompting often degrades LMM performance on perception-heavy tasks, suggesting a potentially harmful overthinking behavior; and *3)* Although the CoT quality is high, LMMs with reflection exhibit significant inefficiency in both normal response and self-correction phases. We hope MME-CoT serves as a foundation for advancing multimodal reasoning in LMMs.

## 1. Introduction

The emergence of Chain-of-Thought (CoT) (Wei et al., 2022) in Large Language Models (LLMs) has demonstrated

*Core Contribution †Project Lead ‡Equal Contribution [1]CUHK MMLab [2]CUHK MiuLar Lab [3]ByteDance [4]NEU [5]UPenn [6]NJU [7]CUHKSZ [8]Shanghai AI Laboratory [9]CPII under InnoHK. Correspondence to: Hongsheng Li <hsli@ee.cuhk.edu.hk>.

*Proceedings of the $42^{nd}$ International Conference on Machine Learning*, Vancouver, Canada. PMLR 267, 2025. Copyright 2025 by the author(s).

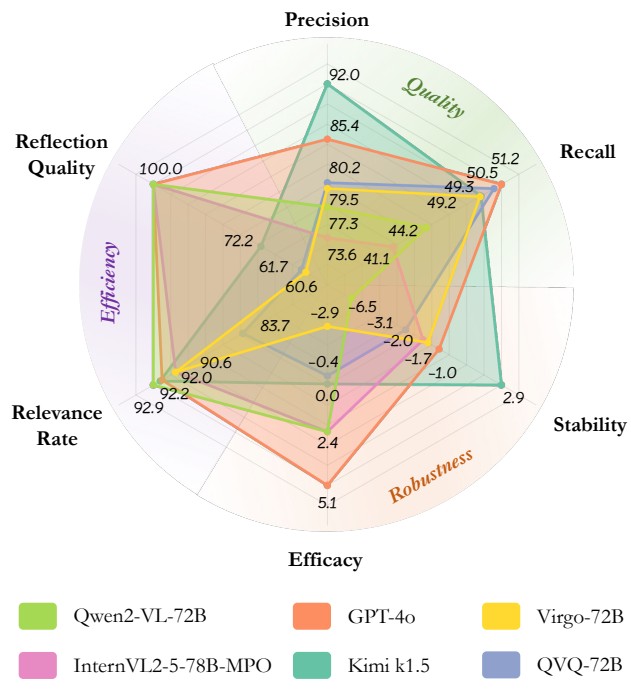

*Figure 1.* **Chain-of-Thought Performance of Leading LMMs in MME-CoT.** Our evaluation suite assesses LMMs using three novel metrics that yield six distinct scores. Results reveal that current open-source models, including those with reflection capabilities, still lag behind closed-source models like GPT-4o and Kimi k1.5 in key aspects of chain-of-thought reasoning.

promising advances in reasoning capabilities, exemplified by the recent OpenAI o1 (OpenAI, 2024a) and DeepSeek-R1 (Guo et al., 2025a). By engaging in a more deliberate, stepwise reasoning process before reaching a final answer, this methodology presents an effective solution in tackling complex scenarios (Lu et al., 2023; Guo et al., 2025c; Jiang et al., 2025; Chen et al., 2025).

In parallel, the multimodal extensions of LLMs, termed Large Multimodal Models (LMMs), have demonstrated remarkable proficiency across diverse visual domains, e.g., general image recognition (Zhang et al., 2023; Zhu et al., 2023; OpenAI, 2023; Zhang et al., 2024a), temporal video

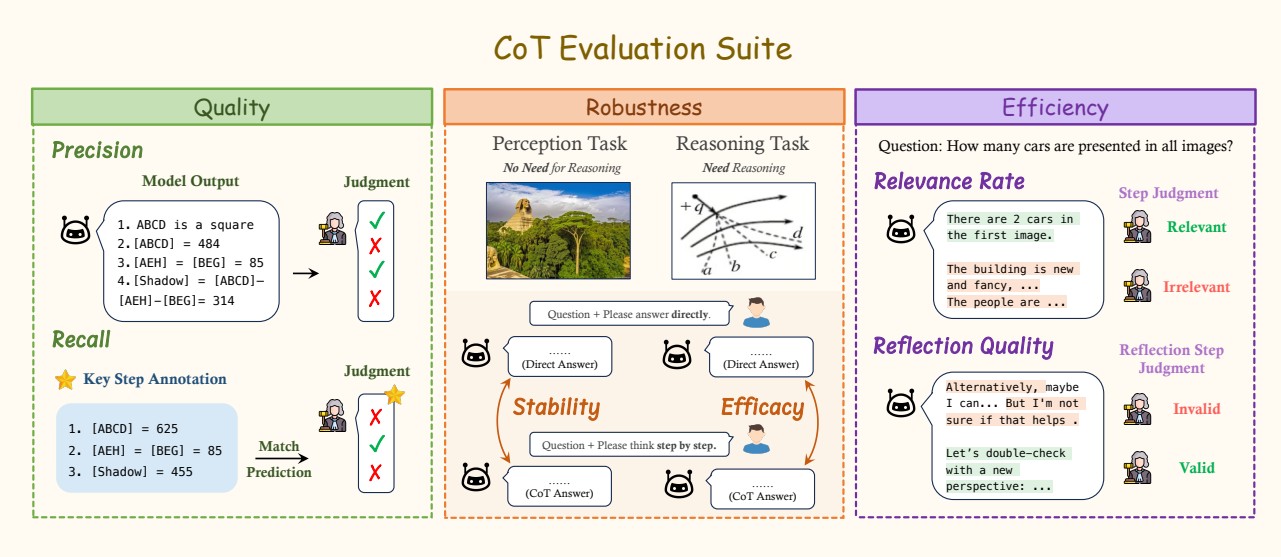

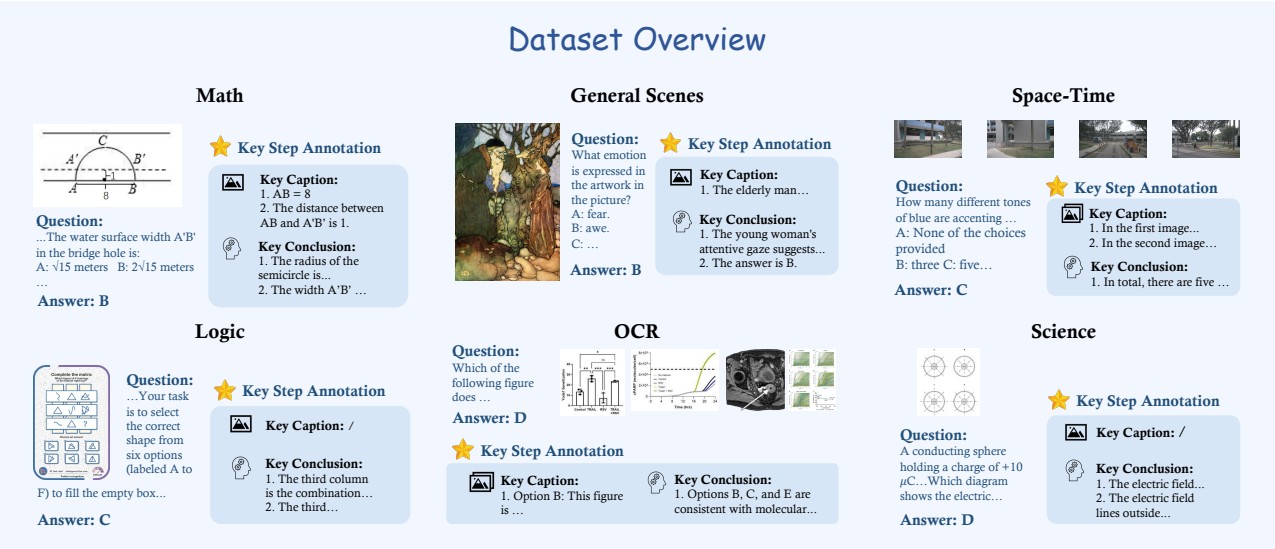

Figure 2. **An Overview of MME-CoT.** Our benchmark contains a comprehensive CoT evaluation suite with three novel aspects and a meticulously curated dataset encompassing six categories.

understanding (Li et al., 2023; Chen et al., 2023), and 3D geometry perception (Guo et al., 2024; Xu et al., 2023; Guo et al., 2023; Jia et al., 2024). However, to what extent and how much CoT reasoning can benefit multimodal challenges still remains an open question. Although some previous efforts (Zhang et al., 2024c; Yu et al., 2023; Zhang et al., 2024d; Guo et al., 2025d) have been made to evaluate the CoT capabilities of LMMs, their examination is insufficiently systematic and thorough, limiting our understanding of multimodal reasoning and its further development.

To bridge this gap, we propose **MME-CoT**, a comprehensive and specialized benchmark for evaluating the CoT rea-

soning skills within LMMs (Figure 2). Our benchmark spans six fundamental domains: math, science, OCR, logic, space-time, and general scenes, encompassing a broad range of CoT-relevant scenarios. Unlike the simplistic metrics used in previous studies, MME-CoT introduces a rigorous evaluation framework that delves into the fine-grained CoT process of LMMs, assessing reasoning quality, robustness, and efficiency. Specifically, we address three critical research questions as follows:

1. ***Is each intermediate CoT step logically valid and faithful without hallucination?*** The outcome-oriented

evaluation paradigm, where most current benchmark adapts, omits the scenario where the model reaches the correct answer through flawed logic or random guess. This causes an illusion of inflated reasoning capabilities in the model. To delve into the reasoning process, we introduce two interpretable metrics to evaluate **the Quality of CoT**: *1) Recall*, which quantifies reasoning informativeness by measuring the proportion of ground-truth solution steps appearing in the response; *2) Precision*, which measures faithfulness by evaluating how many of the generated steps are accurate.

2. ***Does CoT interfere with perception tasks, and to what extent does it enhance reasoning tasks?*** While existing studies primarily focus on the performance improvements CoT brings to reasoning tasks, they often overlook whether CoT could inadvertently disrupt the model's ability to solve perception tasks that require minimal reasoning. To this end, we present *the first* investigation into **the Robustness of CoT** in LMMs. Our benchmark incorporates two task categories (perception and reasoning), and employs two distinct prompting strategies ('direct answer' and 'step-by-step') to assess two metrics: *1) Stability*, which examines whether CoT negatively impacts the model's performance on direct perception tasks; *2) Efficacy*, which measures the extent to which CoT enhances the model's performance on complex reasoning tasks.

3. ***How can we assess the efficiency of CoT in a long reasoning process?*** Recent o1-like models have distinguished themselves by employing excessively long CoT and reflection steps. This raises a critical trade-off question: does this approach strike an optimal balance between accuracy and computational cost? To investigate this, we present *the first* study on **the Efficiency of CoT** in LMMs. We evaluate efficiency using two key metrics: *1) Relevance Rate*, which assesses the proportion of generated content that contributes to answering the question. *2) Reflection Quality*, which analyzes whether each reflection step drives the question towards correctness.

Through our systematic evaluation and analysis, we discover that the fine-grained reflection capability greatly enhances the CoT quality, e.g., QVQ achieves F1 Score of 62.0%, largely surpassing Qwen2-VL-72B by 6.8%. Kimi k1.5 beats GPT-4o and achieves the best quality. As for the robustness, we surprisingly find that most models are interfered with by CoT on the perception tasks, implying a harmful overthinking behavior. The worst case happens in InternVL2.5-8B, where we witness a 6.8% degradation when applying CoT on the perception tasks. This significantly impedes the applicability of models using CoT reasoning as a default practice. Moreover, for CoT efficiency,

we notice that not all steps within the long CoT are related to answering the question, and the model could be distracted by the image content, especially when handling general scenes, space-time, and OCR tasks. Around 30% to 40% of reflection steps fail to help answer questions, pointing out critical issues of current models' reflection capabilities.

The contributions of this paper are summarized as follows:

- The MME-CoT benchmark is curated, covering a comprehensive scope of six multimodal reasoning scenarios. The data collection and annotation process undergoes rigorous human verification, aiming to provide the community with a high-quality evaluation dataset for multimodal reasoning.

- We identify critical issues in existing benchmarks, and introduce a thorough evaluation suite specialized for multimodal CoT reasoning, which meticulously examines the reasoning quality, robustness, and efficiency.

- We conduct extensive experiments and analysis on state-of-the-art LMMs with reasoning capabilities. We summarize our observations and insights, hoping to inspire future advancements of reasoning performance.

Our project page is at https://mmecot.github.io/.

## 2. Dataset Curation

### 2.1. Data Composition and Categorization.

MME-CoT composes 6 major domains with 17 subcategories, as visualized in Fig. 3. Different from textual reasoning questions, the extra visual input significantly enriches the scope of the visual reasoning questions. With the image input, the model needs to frequently visit the image for relevant information according to current reasoning progress. Describing the image area of interest becomes a crucial part of the CoT process. Thus, in addition to complex problems demanding rigorous logic, commonsense scenarios also pose a challenging reasoning problem, as shown in the general scenes in Fig. 2. To maintain focus on the reasoning process, we exclude questions that require complex domain-specific theorems or specialized knowledge.

In addition, to evaluate CoT robustness detailed in Section 3.2, we incorporate a variety of perception tasks along with the reasoning tasks in the benchmark. The reasoning tasks contain questions that demand multi-step logical inference, while the perception tasks consist of questions that primarily test visual recognition abilities or require very minimal reasoning. Existing benchmarks often conflate these two types of tasks, with perception and reasoning questions frequently appearing within the same categories.

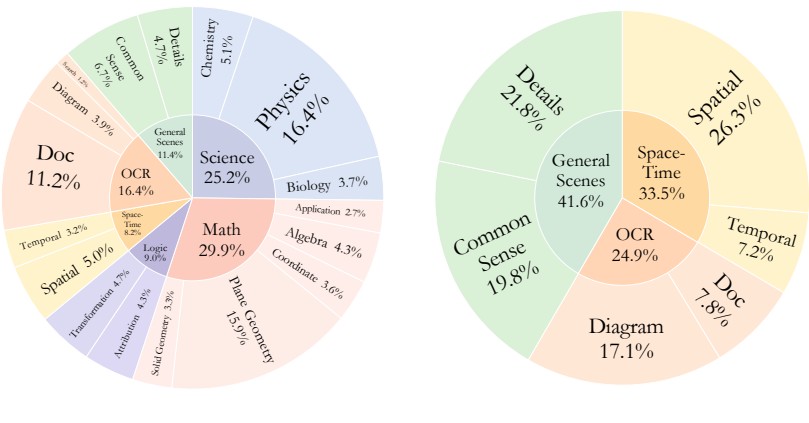

**Reasoning Task**                    **Perception Task**

*Figure 3.* **Category and Subcategory Distribution of MME-CoT.**

| Statistic | Number |
|---|---|
| Total questions | 1,130 |
| - Reasoning questions | 837 (74.1%) |
| Multiple-choice questions | 431 |
| Free-form questions | 406 |
| - Perception questions | 293 (25.9%) |
| Multiple-choice questions | 275 |
| Free-form questions | 18 |
| Total key step annotation | 3,865 |
| - Total inference conclusions | 2,667 |
| - Average inference conclusions | 3.2 |
| - Total image captions | 1,198 |
| - Average image captions | 1.4 |
| Reference image caption item | 1,579 |
| Average reference caption | 1.9 |
| Number of unique images | 2,380 |
| Number of unique questions | 808 |
| Number of unique answers | 271 |
| Maximum question length | 477 |
| Maximum answer length | 15 |
| Average question length | 41.2 |
| Average answer length | 1.2 |

*Table 1.* **Key Statistics of MME-CoT.**

To address this, we implement a two-stage classification approach combining both model-based and human assessment. Initially, we leverage LMMs to guide the preliminary categorization by comparing their performance with and without CoT prompting. We employ GPT-4o (OpenAI, 2024b) and Qwen2-VL-7B (Wang et al., 2024b) to answer questions using both direct and CoT approaches. Superior performance with CoT indicates a reasoning-dominant subcategory, while comparable or inferior CoT performance suggests either perception-focused content or insufficient model reasoning capabilities. The results are shown in Appendix C.2. Subsequently, expert annotators review individual questions to finalize their classification. In total, MME-CoT contains 1,130 questions with 3,865 key step annotation. The detailed statistics of data compositions are shown in Table 1. Please refer to Appendix C for more details about the distribution of data sources.

## 2.2. Data Annotation and Review

To facilitate CoT evaluation, we provide key steps annotation and reference image captions for all the reasoning questions. Key steps are defined as necessary steps for answering the question correctly, falling into two categories: (1) *inference conclusions* - necessary conclusions reached through logical inference steps (including the final answer), and (2) *image captions* - identifications of critical visual information. For efficient annotation, we implement a two-phase process. First, GPT-4o generates initial versions of key steps annotations with questions, images, and ground truth answers as inputs, which yields more accurate rationales compared to question-only prompting. Second, human annotators review these initial versions, correcting any errors or developing key steps independently when GPT-4o fails to provide reasonable output. All steps are reduced to the simplest form, retaining only core conclusions and relevant visual element descriptions. For problems with

multiple solution paths, annotators are required to provide all possible methods. Reference image captions capture visual information not covered by the image captions in key steps, mainly for precision calculation. We use the same two-phase method to obtain these annotations, with annotators verifying and correcting the details.

## 3. CoT Evaluation Strategy

Existing benchmarks only focus on evaluating the final answer of the questions, leaving the whole chain of thoughts unvisited. We argue that the CoT process reflects reasoning capability from multiple aspects, serving as a crucial medium to understand LMM's thinking pattern and deficiency. Here, we present the first holistic CoT evaluation suite to facilitate a comprehensive understanding of the LMMs' reasoning abilities. We detail the evaluation of correctness in Section 3.1, stability and efficacy in Section 3.2, and reflection quality in Section 3.3.

### 3.1. CoT Quality Evaluation

Existing methods typically rely on state-of-the-art LLMs or LMMs to directly evaluate Chain-of-Thought reasoning based on self-defined criteria, using only the final answer as a reference (Hao et al., 2024; Zhang et al., 2024c). We identify two primary issues with the strategy. First, the scoring process only attends to the logical validity of each step, omitting the helpfulness evaluation. Second, there is a large number of complex visual reasoning questions that even the scoring model cannot solve. It is unreasonable for the scoring model to judge another model's reasoning process on these questions without knowing the ground truth solution process. Therefore, building upon our annotated key steps and reference image captions, we leverage two interpretable metrics to evaluate the CoT correctness: recall and precision (Figure 4). The two metrics respectively attend to the

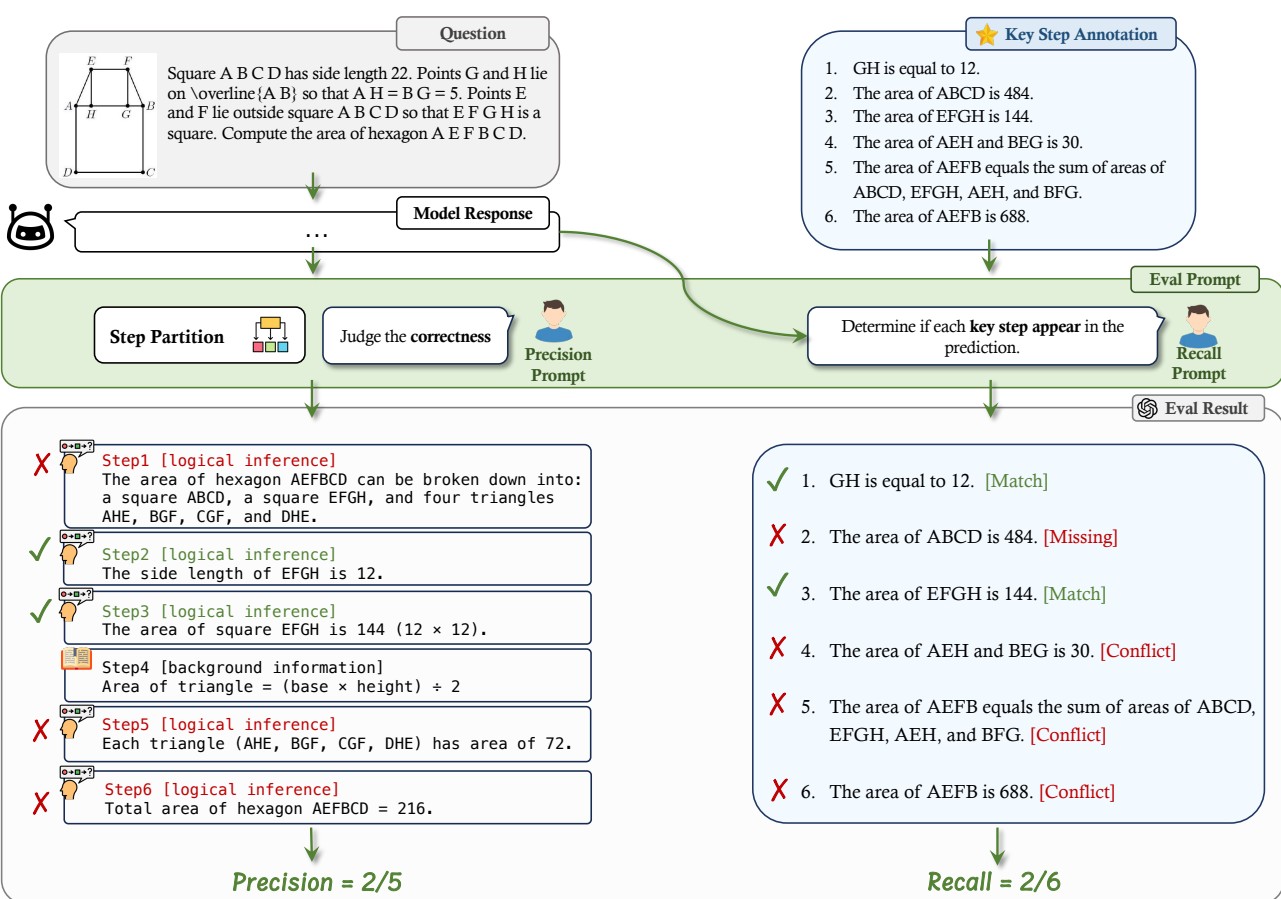

*Figure 4.* **Illustration of CoT Quality Evaluation.** For recall, we prompt GPT-4o to match each key step annotation in the prediction. For precision, GPT-4o is instructed to split the prediction into steps and determine the correctness of all the image caption and logical inference steps.

two aspects of the CoT correctness: informativeness and accuracy. We denote the key steps as $\mathcal{S} = \mathcal{C} \cup \mathcal{I}$, where $\mathcal{C} = \{c_1, ..., c_M\}$ includes $M$ key inference conclusions and $\mathcal{I} = \{i_1, ..., i_N\}$ includes $N$ key image captions.

**Recall.** We prompt GPT-4o (Hurst et al., 2024) to determine whether each key step occurs in the model's CoT response. Then we calculate the ratio of the matched key steps $\mathcal{S}_{\text{matched}}$ against all the annotated key steps:

$$k_0 = \arg\max_k \frac{\left|\mathcal{S}_{\text{matched}}^k\right|}{|\mathcal{S}^k|}, \quad (1)$$

$$\text{Recall}_\mathcal{C} = \frac{\left|\mathcal{C}_{\text{matched}}^{k_0}\right|}{|\mathcal{C}^{k_0}|}, \quad \text{Recall}_\mathcal{I} = \frac{\left|\mathcal{I}_{\text{matched}}^{k_0}\right|}{|\mathcal{I}^{k_0}|}, \quad (2)$$

$$\text{Recall} = \frac{\left|\mathcal{S}_{\text{matched}}^{k_0}\right|}{|\mathcal{S}^{k_0}|}. \quad (3)$$

where $\mathcal{S}^k$ denotes the $k^{\text{th}}$ method of the problem. Intuitively, recall measures how many informative steps are reached

by the model. From another perspective, this metric also strictly examines the process's rigorousness toward reaching the correct answer, eliminating the probability of random guessing. For questions with multiple methods, we compute the recall on the most matched method.

**Precision.** We first instruct GPT-4o to partition the prediction into a sequence of steps $\mathcal{P}$, as shown in Fig. 7 in the Appendix. Each step is categorized into one of three classes: logical inference, image caption, and background information. The logical inference step draws an intermediate or final conclusion based on the previously obtained information. The image caption step depicts elements of interest in the image. The background information step states external knowledge or question information. Visual reasoning can be primarily characterized as an interleaved sequence of image captions and logical inferences, so we focus on measuring precision for these two key step types. We assess the correctness of logical inference steps ($\mathcal{C}^\mathcal{P}$) and image caption steps ($\mathcal{I}^\mathcal{P}$) using two criteria: 1. If the step exists

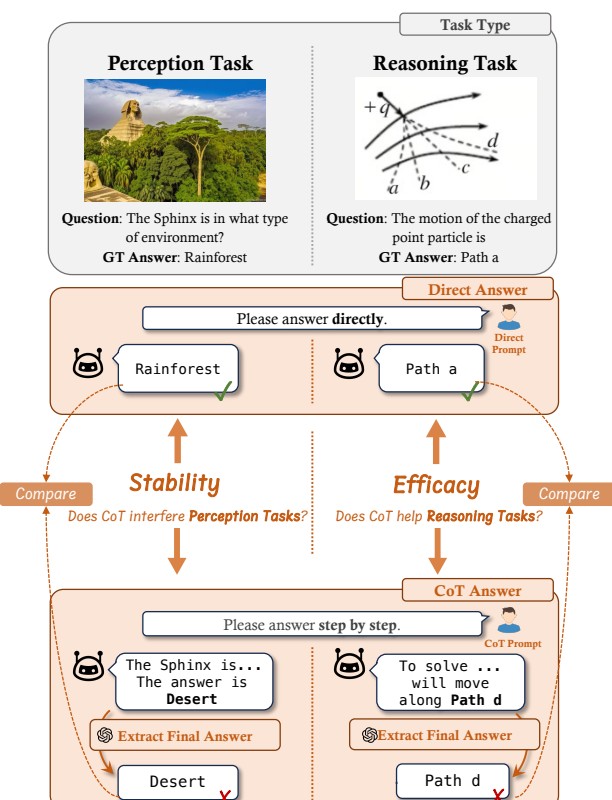

*Figure 5.* **Illustration of CoT Robustness Evaluation.** We compare the performance of applying CoT prompt and direct prompt on two types of tasks: perception and reasoning. The stability score measures whether CoT interferes with perception, while the efficacy score assesses the performance gain of CoT on reasoning tasks.

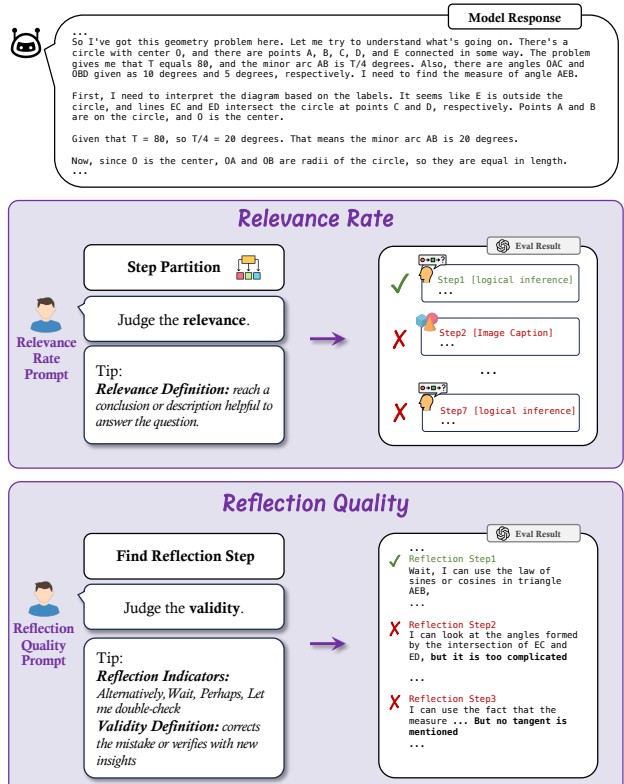

*Figure 6.* **Illustration of CoT Efficiency Evaluation.** For relevance rate, we partition the prediction into steps and determine if it is relevant by GPT-4o. For reflection quality, we prompt GPT-4o to identify the reflection steps by common indicators and judge the validity of the reflection. The definitions of relevance and validity are included.

in $\mathcal{S}$, the step is correct. 2. If the step is logically correct or faithfully depicts the image based on the annotations, the step is also correct. Thus, we compute precision as:

$$\text{Precision}_{\mathcal{C}} = \frac{|\mathcal{C}^{\mathcal{P}}_{\text{correct}}|}{|\mathcal{C}^{\mathcal{P}}|}, \quad \text{Precision}_{\mathcal{I}} = \frac{|\mathcal{I}^{\mathcal{P}}_{\text{correct}}|}{|\mathcal{I}^{\mathcal{P}}|}, \quad (4)$$

$$\text{Precision} = \frac{|\mathcal{C}^{\mathcal{P}}_{\text{correct}} \cup \mathcal{I}^{\mathcal{P}}_{\text{correct}}|}{|\mathcal{C}^{\mathcal{P}} \cup \mathcal{I}^{\mathcal{P}}|} \quad (5)$$

Intuitively, precision evaluates the faithfulness of each step, considering all the possible reasoning output. Finally, we calculate the F1 score as the metric of CoT quality.

### 3.2. CoT Robustness Evaluation

Here, we perform the first investigation on the robustness of CoT in visual reasoning. The effectiveness of CoT on reasoning tasks has been verified in many works (Wei et al., 2022; OpenAI, 2024a). However, how CoT impacts visual perception tasks or tasks requiring minimal reasoning still remains unknown. Despite the neglect, this question bears

great importance. In real-world applications, what task is given is unknown in advance. Whether the model should perform CoT to solve the task is difficult to determine. In fact, there exists no golden standard to determine which question can benefit from CoT so far (Sprague et al., 2024). Instead of trying to define this criterion, we examine the performance of CoT across all kinds of tasks, both reasoning and perception. We argue that an ideal CoT process should assist in reasoning and not interfere with pure perception. Therefore, it can be applied for any tasks. Based on this, we propose to evaluate two metrics of CoT: stability and efficacy (Figure 5). We leverage two kinds of prompts: the direct prompt (DIR) and the CoT prompt (COT). The direct prompt asks the model to directly provide the final answer, while the CoT prompt instructs the model to perform step-by-step reasoning and finally give the answer. To directly compare the performance difference caused by these two prompts, we conduct the direct evaluation, which only judges the correctness of the final answer, i.e., accuracy. We instruct GPT-4o mini (Hurst et al., 2024) to extract the final answer, and then compare it with the ground truth answer,

following the two-step procedure introduced in (Zhang et al., 2024c).

**Stability.** We define the performance difference of the two prompts on the perception tasks $\mathbf{P}$ as the stability score:

$$\text{Stability} = \text{Acc}_{\text{COT}}^{\mathbf{P}} - \text{Acc}_{\text{DIR}}^{\mathbf{P}}. \tag{6}$$

Intuitively, applying the CoT prompt to perception tasks should not degrade performance compared with the direct prompt. Thus, a model with stable CoT should be not less than 0. Otherwise, the model's thinking process demonstrates inconsistency and harm. The overthinking process pushes over the original correct judgment.

**Efficacy.** Similarly, the performance difference of the two prompts on the reasoning tasks $\mathbf{R}$ is defined as the score:

$$\text{Efficacy} = \text{Acc}_{\text{COT}}^{\mathbf{R}} - \text{Acc}_{\text{DIR}}^{\mathbf{R}}. \tag{7}$$

Intuitively, CoT facilitates stepwise thinking and therefore benefits answering reasoning tasks. The difference reflects how much CoT can enhance reasoning.

### 3.3. CoT Efficiency Evaluation

Models like o1 generate extremely long thinking processes with reflection and verification of current steps and outcomes. We perform the first exhaustive analysis of the CoT efficiency of visual reasoning with two carefully designed metrics (Figure 6):

**Relevance Rate.** Although the long reasoning content allows for deeper thinking, it may also introduce a large amount of irrelevant information. As shown in the bottom left of Fig. 6, the model has identified the critical element in the image for answering the question, but it still generates a detailed description of other objects. This irrelevant information provides no helpful information to work out the answer. In the meantime, this extra content slows down the generation speed. Similar to the calculation of precision, we employ the same method to partition the prediction into steps. Then, we instruct GPT-4o to determine all the relevant steps $\mathcal{P}_{\text{relevant}}$. The step is considered relevant only when the majority of its content works towards solving the question. We first compute the raw relevance rate and then apply a scaling factor to amplify the differences between models. Let $r_x$ denote the raw relevance rate:

$$r_{\mathcal{C}} = \frac{|\mathcal{C}_{\text{relevant}}^{\mathcal{P}}|}{|\mathcal{C}^{\mathcal{P}}|}, \quad r_{\mathcal{I}} = \frac{|\mathcal{I}_{\text{relevant}}^{\mathcal{P}}|}{|\mathcal{I}^{\mathcal{P}}|}, \tag{8}$$

$$r = \frac{|\mathcal{P}_{\text{relevant}}|}{|\mathcal{P}|}. \tag{9}$$

Then, the final relevance rate Relevance Rate$_x$ is defined as:

$$\text{Relevance Rate}_x = \frac{r_x - \alpha}{1 - \alpha}, \quad x \in \mathcal{C}, \mathcal{I}, \emptyset \tag{10}$$

where $x = \emptyset$ corresponds to the overall relevance rate, and we take $\alpha$ as 0.8.

**Reflection Quality.** The superior reasoning ability could be largely attributed to the reflection and verification process. However, our analysis reveals that not all reflective steps contribute meaningfully to finding correct answers. We identify distinct failure patterns in the reflection process. Some reflective steps mislead the reasoning by introducing new errors or incorrect assumptions, while others are redundant, simply echoing previous conclusions without contributing new insights. To account for failure reflection scenarios, we propose to measure the validity of the reflection. We define a valid reflection as either correctly pointing out the previous mistakes or verifying the previous conclusion with a new insight. Otherwise, the reflection only slows down the reasoning. To instruct GPT-4o to determine all the valid reflection steps $\mathcal{R}$, we list a set of common indicators of the start of the reflection, such as "Wait" and "Alternatively", and illustrate the definition of valid reflection. For all the valid reflection steps $\mathcal{R}_{\text{valid}}$, the reflection quality is computed as:

$$\text{Reflection Quality} = \frac{|\mathcal{R}_{\text{valid}}|}{|\mathcal{R}|}. \tag{11}$$

## 4. Experiments

In this section, we conduct a systematic evaluation of state-of-the-art models on MME-CoT. We first detail the experiment setup in Section 4.1. Then in Section 4.2, we report the quantitative results and provide valuable insights derived from our analysis.

### 4.1. Experiment Setup

**Evaluation Models.** We select top-performing LMMs for comprehensive CoT evaluation. We test earlier models such as LLaVA-OneVision (7B, 72B) (Li et al., 2024a), Qwen2-VL (7B, 72B) (Qwen Team, 2024), MiniCPM-V-2.6 (Yao et al., 2024b), and InternVL2.5 (8B) (Chen et al., 2024c), which are not trained for the reasoning capability. We also include GPT-4o (OpenAI, 2024b) as a strong baseline model. Besides, we test recent models targeting reasoning, including LLaVA-CoT (11B) (Xu et al., 2024), Mulberry (8B) (Yao et al., 2024a), InternVL2.5-MPO (8B, 78B) (Wang et al., 2024c). Finally, we evaluate LMMs with reflection capabilities, including both closed-source models like Kimi k1.5 (Team et al., 2025) and open-source implementations such as QVQ-72B (Team, 2024) and Virgo-72B (Du et al., 2025).

**Implementation Details.** We define the CoT prompt as: *Please generate a step-by-step answer, include all your intermediate reasoning process, and provide the final answer*

*Table 2.* **Evaluation Results of Three Aspects of CoT in MME-CoT.** Models with † receive detailed image captions instead of images. We mark the highest score of each metric in red. ∗ denotes unreliable results due to the refusal to answer directly most of the time.

| Model | CoT Quality | | | | | | | CoT Robustness | | | | | | | CoT Efficiency | | | | |
|---|---|---|---|---|---|---|---|---|---|---|---|---|---|---|---|---|---|---|---|
| | F1 Score | Precision | Image | Conclusion | Recall | Image | Conclusion | Avg. Score | Stability | CoT Perception | Direct Perception | Efficacy | CoT Reasoning | Direct Reasoning | Avg. Score | Relevance Rate | Image | Conclusion | Reflection Quality |
| *Open-source LMMs* | | | | | | | | | | | | | | | | | | | |
| Mulberry | 27.4 | 59.1 | 74.1 | 53.8 | 17.8 | 26.5 | 17.1 | 2.0* | 5.8* | 35.7* | 29.9* | -1.9* | 8.5* | 10.4* | 89.5 | 79.0 | 50.8 | 95.4 | 100 |
| LLaVA-OV-7B | 30.9 | 50.9 | 47.2 | 43.5 | 22.2 | 24.4 | 23.2 | -3.4 | -3.8 | 46.1 | 49.8 | -3.0 | 16.4 | 19.4 | 96.7 | 93.4 | 85.3 | 93.4 | 100 |
| LLaVA-CoT | 34.9 | 53.9 | 75.6 | 46.2 | 25.8 | 35.8 | 24.4 | - | - | 51.5 | - | - | 24.4 | - | 99.4 | 98.8 | 85.2 | 95.4 | 100 |
| DeepSeek-VL2 | 34.9 | 54.4 | 44.5 | 46.3 | 25.7 | 33.4 | 26.0 | -3.3 | -5.1 | 60.8 | 65.9 | -1.6 | 27.4 | 29.0 | 96.7 | 93.4 | 80.8 | 99.0 | 100 |
| LLaVA-OV-72B | 36.3 | 57.3 | 43.4 | 50.6 | 26.6 | 29.5 | 27.4 | -0.2 | 0.3 | 61.1 | 60.8 | -0.6 | 27.6 | 28.2 | 99.1 | 98.1 | 89.6 | 97.7 | 100 |
| MiniCPM-V-2.6 | 39.8 | 57.3 | 63.4 | 45.4 | 30.5 | 47.5 | 26.7 | -3.3 | -4.5 | 59.6 | 64.0 | -2.2 | 26.1 | 28.3 | 97.8 | 85.7 | 74.6 | 97.6 | 100 |
| InternVL2.5-8B | 41.1 | 60.0 | 52.4 | 50.8 | 31.3 | 40.4 | 30.6 | -3.0 | -6.8 | 56.8 | 63.8 | 0.9 | 30.4 | 29.5 | 98.4 | 96.8 | 93.0 | 98.9 | 100 |
| Qwen2-VL-7B | 42.1 | 61.6 | 61.0 | 49.3 | 32.0 | 46.6 | 30.5 | -3.7 | -2.8 | 59.8 | 62.6 | -4.6 | 25.8 | 30.4 | 94.9 | 89.8 | 80.3 | 98.8 | 100 |
| InternVL2.5-8B-MPO | 43.0 | 60.4 | 60.8 | 49.9 | 33.4 | 44.9 | 31.8 | 0.6 | 0.3 | 62.5 | 62.1 | 0.8 | 30.1 | 29.3 | 94.7 | 89.3 | 84.0 | 96.4 | 100 |
| InternVL2.5-78B-MPO | 52.7 | 73.6 | 68.4 | 63.0 | 41.1 | 53.6 | 39.1 | 0.2 | -2.0 | 68.3 | 70.3 | 2.4 | 38.0 | 35.6 | 99.6 | 99.2 | 98.1 | 98.0 | 100 |
| Qwen2-VL-72B | 56.2 | 77.3 | 67.2 | 70.3 | 44.2 | 57.1 | 42.2 | -2.0 | -6.3 | 68.9 | 75.2 | 2.3 | 38.9 | 36.6 | 96.5 | 92.9 | 86.0 | 98.7 | 100 |
| Virgo-72B | 60.8 | 79.5 | 71.6 | 72.7 | 49.2 | 60.5 | 47.7 | -2.6* | -2.2* | 73.7* | 75.9* | -3.1* | 46.4* | 49.5* | 75.0 | 90.6 | 79.8 | 95.6 | 60.6 |
| QVQ-72B | 62.0 | 80.2 | 73.9 | 77.5 | 50.5 | 60.1 | 48.9 | - | - | 70.0 | - | - | 41.0 | - | 74.0 | 83.7 | 63.9 | 95.1 | 61.7 |
| *Closed-source LMMs* | | | | | | | | | | | | | | | | | | | |
| Claude-3.5 | 59.4 | 77.2 | 65.2 | 71.6 | 48.2 | 59.7 | 47.6 | 10.5 | 11.0 | 74.1 | 63.1 | 9.9 | 41.5 | 31.6 | 95.4 | 90.9 | 79.5 | 99.0 | 100 |
| Gemini-2.0-Flash | 63.8 | 80.3 | 60.2 | 74.6 | 52.9 | 56.7 | 53.6 | 6.3 | 5.9 | 78.4 | 72.5 | 6.6 | 47.5 | 40.9 | 97.7 | 95.5 | 91.4 | 98.4 | 100 |
| GPT-4o | 64.0 | 85.4 | 73.3 | 81.4 | 51.2 | 64.3 | 49.9 | 2.1 | -1.0 | 71.0 | 72.0 | 5.1 | 40.6 | 35.5 | 96.0 | 92.0 | 82.4 | 99.1 | 100 |
| Kimi k1.5 | 64.2 | 92.0 | 78.1 | 89.8 | 49.3 | 62.9 | 47.9 | 1.4 | 2.9 | 65.7 | 62.9 | 0.0 | 40.0 | 40.0 | 82.2 | 92.2 | 82.2 | 97.2 | 72.2 |

*at the end.* and the direct prompt as: *Please directly provide the final answer without any other output.* We only calculate recall of image observation and logical inference on questions where key inference conclusion or image observation exists. We employ GPT-4o mini for the direct evaluation and GPT-4o for all other criteria. For hyperparameters, we follow the settings in VLMEvalKit (Duan et al., 2024).

### 4.2. Quantitative Results

We conduct extensive experiments on various LMMs with our proposed CoT evaluation suite. The main results are presented in Table 2 and Table 3. We begin by analyzing the overall performance and then highlight key findings.

**Overall Results.** In Table 2, we present the overall performance of three CoT evaluation perspectives with specific metrics. To provide a comprehensive understanding, we report precision, recall, and relevance for both logical inference and image caption steps. For robustness, we provide the direct evaluation result on the perception and reasoning tasks, with either CoT or direct prompt. We employ the average value of the stability and efficacy as the final robustness metric. Notably, we define the reflection quality as 100 on models incapable of reflection.

For CoT quality, Kimi k1.5 achieves the highest F1 score. Open-source models with larger sizes consistently demonstrate better performance, highlighting the scalability of LMMs. Notably, Qwen2-VL-72B outperforms all other open-source models without reflection, even surpassing InternVL2.5-78B-MPO, which is specifically enhanced for reasoning. Analysis reveals that GPT-4o achieves superior performance across all recall metrics, while Kimi k1.5

demonstrates the highest scores in precision evaluations. For CoT robustness, Mulberry obtains the highest average score. However, when we look into its output, we find it still generates lengthy rationales despite receiving a direct prompt. Even worse, the direct prompt seems to be an out-of-distribution input for Mulberry, frequently leading to nonsensical outputs. Further analysis of other models' predictions reveals that LLaVA-CoT, Virgo, QVQ, and Kimi k1.5 similarly neglect the direct prompt, instead generating extended rationales before answering. Consequently, their robustness scores may be misleading. Once again, GPT-4o achieves the highest robustness score. Among open-source models, only InternVL2.5-MPO, in both its 8B and 78B variants, attains a positive robustness score. Finally, for CoT efficiency, InternVL2.5-8B obtains the maximum relevance of 98.4%, suggesting its consistent focus on questions.

Now, we summarize our key observations as follows:

***Models with reflection largely benefit CoT quality.*** As shown in Table 2, the F1 scores of the two models with reflection capability most closely approach GPT-4o. After specifically fine-tuning for the reasoning capabilities from Qwen2-VL-72B, QVQ surpasses its base model by 5.8%. Notably, although QVQ generates longer CoT sequences than Qwen2-VL-72B, QVQ's precision still exceeds Qwen2-VL-72B by 2.9%, indicating superior accuracy in each reasoning step. Kimi k1.5 also surpasses the previous state-of-the-art model GPT-4o, obtaining the highest CoT quality.

***Long CoT does not necessarily cover key steps.*** Despite high precision in long CoT models, the informativeness of each step is not guaranteed. We observe that the recall trend

*Table 3.* **Evaluation Results of Three Aspects of CoT across Categories in MME-CoT.** Models with † receive detailed image captions instead of images. Best performance is marked in red. ∗ denotes unreliable results due to the refusal to answer directly most of the time.

| Model | General Scenes | | | Space-Time | | | OCR | | | Math | | Science | | Logic | |
|---|---|---|---|---|---|---|---|---|---|---|---|---|---|---|---|
| | Quality | Robustness | Efficiency | Quality | Robustness | Efficiency | Quality | Robustness | Efficiency | Quality | Efficiency | Quality | Efficiency | Quality | Efficiency |
| *Open-source LMMs* | | | | | | | | | | | | | | | |
| Mulberry | 33.9 | 4.3* | 88.0 | 18.2 | 1.0* | 69.2 | 26.7 | 6.6* | 63.2 | 29.1 | 93.9 | 29.1 | 96.1 | 13.9 | 99.5 |
| LLaVA-OV-7B | 41.8 | -6.2 | 90.9 | 23.8 | -6.7 | 62.4 | 44.1 | -0.2 | 71.4 | 27.4 | 98.7 | 28.5 | 97.9 | 12.2 | 99.0 |
| LLaVA-CoT | 38.2 | - | 94.9 | 33.6 | - | 84.5 | 37.4 | - | 88.9 | 35.3 | 95.5 | 36.4 | 96.7 | 14.9 | 98.6 |
| DeepSeek-VL2 | 43.9 | -2.3 | 94.0 | 43.4 | -11.5 | 93.2 | 54.1 | -2.4 | 92.8 | 24.5 | 99.8 | 25.2 | 98.1 | 30.0 | 94.0 |
| LLaVA-OV-72B | 41.8 | -2.3 | 99.5 | 29.0 | -0.9 | 71.8 | 40.8 | -1.7 | 92.0 | 38.4 | 99.4 | 35.4 | 97.8 | 18.4 | 91.1 |
| MiniCPM-V-2.6 | 47.1 | 3.2 | 93.9 | 49.3 | -14.4 | 85.5 | 63.7 | -4.9 | 81.0 | 32.9 | 97.6 | 29.5 | 95.2 | 16.9 | 96.8 |
| InternVL2.5-8B | 43.8 | -6.4 | 93.6 | 50.7 | -8.9 | 99.5 | 44.7 | -4.1 | 99.4 | 40.9 | 99.0 | 40.8 | 98.6 | 19.5 | 98.4 |
| Qwen2-VL-7B | 46.7 | -3.4 | 89.6 | 51.7 | -11.8 | 86.5 | 65.9 | 0.9 | 93.1 | 34.0 | 98.9 | 34.6 | 98.6 | 18.4 | 88.4 |
| InternVL2.5-8B-MPO | 47.2 | 2.9 | 97.1 | 51.8 | -0.2 | 87.3 | 59.6 | -1.0 | 90.7 | 37.4 | 96.7 | 39.0 | 97.8 | 20.9 | 89.9 |
| InternVL2.5-78B-MPO | 47.9 | 0.0 | 94.6 | 55.5 | -2.3 | 95.9 | 72.2 | 2.7 | 86.6 | 50.6 | 98.9 | 48.5 | 98.9 | 24.2 | 93.6 |
| Qwen2-VL-72B | 51.9 | -2.9 | 94.4 | 59.7 | -5.3 | 93.4 | 77.6 | 2.5 | 90.9 | 49.6 | 98.9 | 53.6 | 99.5 | 40.0 | 94.0 |
| Virgo-72B | 60.5 | 0.5* | 71.9 | 59.6 | -3.8* | 65.1 | 79.9 | -1.0* | 75.6 | 59.6 | 80.2 | 55.5 | 81.0 | 39.6 | 59.3 |
| QVQ-72B | 62.6 | - | 70.1 | 58.2 | - | 51.4 | 76.9 | - | 60.7 | 61.4 | 81.1 | 57.7 | 83.2 | 44.6 | 61.1 |
| *Closed-source LMMs* | | | | | | | | | | | | | | | |
| Claude-3.5 | 63.8 | 7.1 | 95.4 | 51.7 | 11.4 | 83.4 | 65.8 | 10.1 | 91.1 | 53.0 | 98.9 | 59.6 | 98.6 | 41.7 | 88.4 |
| Gemini-2.0-Flash | 55.5 | 7.2 | 95.0 | 56.1 | 7.3 | 95.7 | 81.5 | 3.2 | 96.6 | 65.1 | 97.0 | 65.9 | 99.1 | 34.8 | 99.8 |
| GPT4o | 62.3 | -1.7 | 98.1 | 66.3 | 5.5 | 82.3 | 83.3 | -1.0 | 91.0 | 60.8 | 98.4 | 64.1 | 99.7 | 27.2 | 96.0 |
| Kimi k1.5 | 73.8 | 14.0 | 84.1 | 69.3 | -3.3 | 58.9 | 85.3 | 7.5 | 83.6 | 58.3 | 90.8 | 48.8 | 85.9 | 41.3 | 60.3 |

among GPT-4o, QVQ, and Virgo does not align with their CoT Rea. performance (i.e., their final answer accuracy on the reasoning tasks under the CoT prompt). Specifically, while both Virgo and QVQ outperform GPT-4o in direct evaluation, they lag behind in recall. This suggests that long CoT models sometimes reach correct answers while skipping intermediate steps, which contradicts the principle of stepwise reasoning and warrants further investigation.

***CoT impairs perception task performance in most models.*** Surprisingly, most models exhibit negative stability scores, indicating that CoT interferes with perception tasks. The most significant degradation occurs in InternVL2.5-8B, where performance drops by 6.8%. This reveals inconsistency and potential overthinking in current models, presenting a significant barrier to adopting CoT as the default answering strategy. Among models that provide direct answers, only LLaVA-OV-72B and InternVL2.5-8B-MPO achieve a modest positive score of 0.3%.

***More parameters enable models to grasp reasoning better.*** We find that models with larger parameter counts tend to achieve higher efficacy scores. This pattern is evident across LLaVA-OV, InternVL2.5-MPO, and Qwen2-VL. For instance, while Qwen2-VL-7B shows a 4.8% decrease in performance when applying CoT to reasoning tasks, its larger counterpart, Qwen2-VL-72B, demonstrates a 2.4% improvement. This discrepancy suggests that models with more parameters could better grasp the reasoning ability under the same training paradigm.

***Reflection often fails to help.*** While reflection is a key feature of long CoT models for answer verification, both QVQ and Virgo achieve reflection quality scores of only about 60%, indicating that approximately 40% of reflection attempts fail to contribute meaningfully to answer accuracy. Even for the closed-source model Kimi k1.5, over 25% reflection steps are also invalid. This substantial failure rate compromises efficiency by potentially introducing unnecessary or distracting steps before reaching correct solutions. Future research should explore methods to reduce these ineffective reflections to improve both efficiency and quality.

## 5. Conclusion

In this paper, we have introduced MME-CoT, a comprehensive benchmark designed to evaluate Chain-of-Thought reasoning in Large Multimodal Models. Our dataset comprises six categories to cover most scenarios of visual reasoning tasks. To gain a thorough understanding of the reasoning process, we design a novel CoT evaluation suite with three metrics. Our systematic evaluation obtains useful insights into the issues within the current state-of-the-art Large Multimodal Models. We identify critical flaws in all the tested open-source models. As the field continues to evolve, MME-CoT stands as a valuable tool for measuring progress and identifying areas for improvement in the development of more sophisticated multimodal AI systems.

## Acknowledgements

This project is supported in part by the Centre for Perceptual and Interactive Intelligence (CPII) Ltd under the Innovation and Technology Commission (ITC)'s InnoHK, and in part by NSFC-RGC Project N_CUHK498/24. Hongsheng Li is a PI of CPII under the InnoHK.

## Impact Statement

This paper presents work whose goal is to advance the field of Computer Vision and Machine Learning. There are many potential societal consequences of our work, none of which we feel must be specifically highlighted here.

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

## Appendix Overview

## A. Related Work

### A.1. Large Multimodal Models

The field of multimodal (Radford et al., 2021; Li et al., 2022; OpenAI, 2023; Rombach et al., 2022; Shen et al., 2025; Jiang et al., 2024a; Zong et al., 2024a) AI has experienced extraordinary growth, particularly through the development of Large Multimodal Models (LMMs) (Liu et al., 2023; Zhu et al., 2023; Lin et al., 2023; Zong et al., 2024b; Qwen Team, 2024). These models build upon the achievements of Large Language Models (LLMs) (Touvron et al., 2023; Yang et al., 2024) and advanced vision models (Radford et al., 2021), expanding their capabilities to process multiple kinds of visual input (Li et al., 2024b; Guo et al., 2023; Li et al., 2023; Guo et al., 2025b).

Closed-source models, such as OpenAI's GPT-4o (OpenAI, 2024b), have demonstrated exceptional capabilities in visual understanding and reasoning. However, their closed-source nature creates barriers to widespread adoption and further development by the broader research community. In response, significant progress has been made in developing open-source alternatives. Early approaches like LLaVA (Liu et al., 2023), LLaMA-Adapter (Zhang et al., 2024b), and MiniGPT-4 (Zhu et al., 2023) established a foundation by combining frozen CLIP models for image encoding with LLMs, enabling multimodal instruction tuning. Subsequent developments through projects such as InternVL2 (Chen et al., 2024d), Qwen2-VL (Qwen Team, 2024), SPHINX (Gao et al., 2024; Lin et al., 2023), and MiniCPM-V (Yao et al., 2024b) have expanded these capabilities by incorporating more diverse visual instruction datasets and broadening application scenarios.

Recently, with the introduction of o1 (OpenAI, 2024a), the field of LMMs has also focused on enhancing the reasoning capability. (Wang et al., 2024d) introduces mixed preference optimization with automatically constructed data. (Yao et al., 2024a) proposes to leverage collective knowledge from multiple models to identify effective reasoning paths. Besides, several works (Team, 2024; Du et al., 2025) have demonstrated the ability to replicate behaviors similar to o1 models, particularly regarding multi-step CoT reasoning with iterative self-reflection and verification processes.

### A.2. Reasoning Evaluation

Several methods have been developed to evaluate reasoning in natural language processing, including ROSCOE (Golovneva et al., 2022) and ReCEval (Prasad et al., 2023), which assess reasoning chains across multiple dimensions such as correctness and informativeness. However, these approaches are limited to text-only scenarios and do not address the unique challenges present in visual reasoning tasks. Furthermore, the emergence of long chain-of-thought (CoT) reasoning has introduced additional considerations, such as output efficiency and reflection quality, which existing evaluation methods do not adequately address.

On the other hand, various multimodal benchmarks have been developed to assess reasoning abilities across specific domains. Current exploration of visual reasoning predominantly focuses on the mathematics (Zhang et al., 2024d; Peng et al., 2024) domains. MathVista (Lu et al., 2023) provides a comprehensive collection of mathematical problems that assess mathematical and logical reasoning abilities. Building on this, MathVerse (Zhang et al., 2024c) introduces a new benchmark by eliminating redundant textual information to evaluate whether LMMs can accurately interpret graphical representations. OlympiadBench (He et al., 2024) further raises the complexity bar by incorporating challenging Olympiad-level mathematics and physics problems. Despite these advances in specialized domains, broader applications such as general-scene reasoning remain relatively unexplored. Recent developments have begun to expand beyond purely scientific reasoning. For instance,

M³CoT (Chen et al., 2024b) and SciVerse (Guo et al., 2025c) incorporate commonsense tasks alongside scientific reasoning and knowledge-based assessment in the multimodal benchmark. However, most existing benchmarks focus solely on evaluating final answers while overlooking the intermediate steps, thus providing limited insights into the process through which models arrive at their conclusions.

## B. More Evaluation Details

We provide the illustration of the step partition in Figure 7.

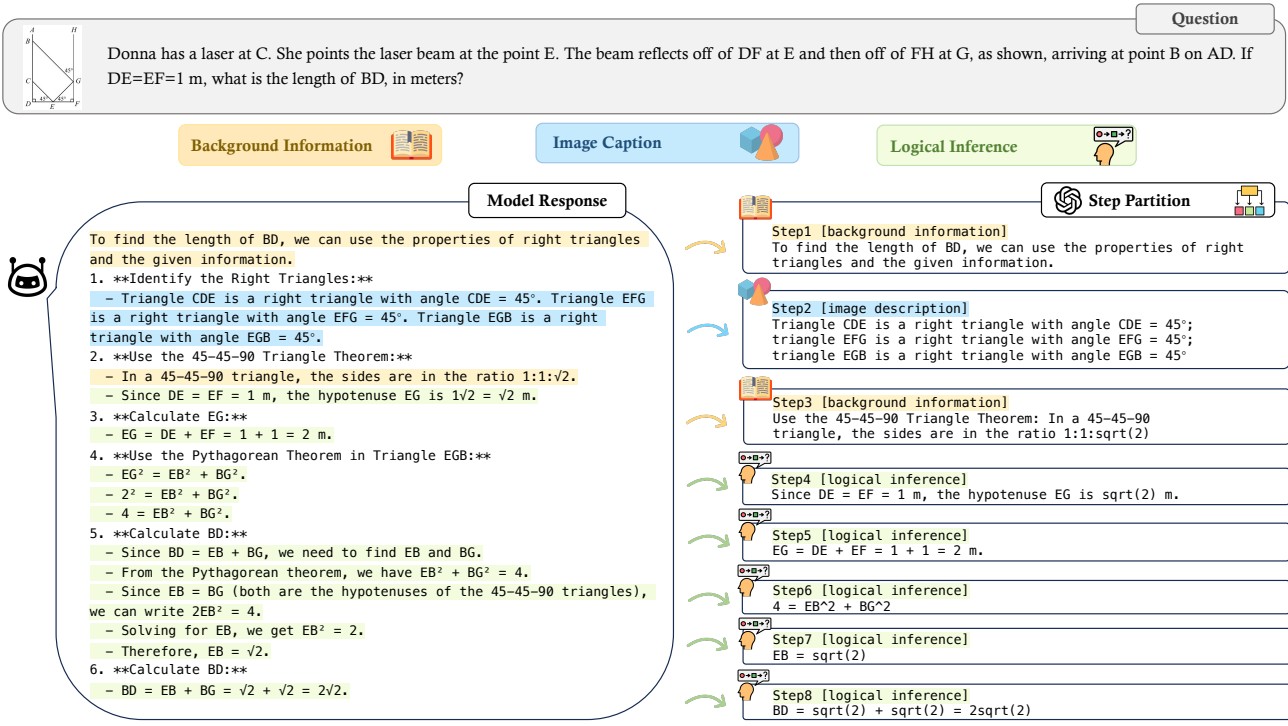

*Figure 7.* **Illustration of Step Partition.** We instruct GPT-4o to divide each step into three categories: image caption, background information, or logical inference. The step partition result is later used to perform step-wise reasoning evaluation. We focus on evaluating the image caption and logical inference steps, which are the keys to visual reasoning.

## C. More Dataset Details

### C.1. Data Source Distribution

We visualize the data source distributions in our benchmark, which consists of 15 sets, including MathVerse (Zhang et al., 2024c), MMMUPro (Yue et al., 2024), OlympiadBench (He et al., 2024), MMT-Bench (Ying et al., 2024), MuirBench (Wang et al., 2024a), ml-rpm-bench (Zhang et al., 2024e), MMSearch (Jiang et al., 2024b), CharXiv (Wang et al., 2024e), and SciVerse (Guo et al., 2025c).

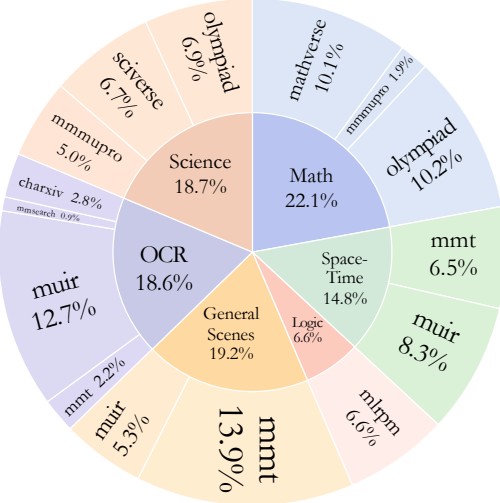

*Figure 8.* **Data Source Distribution of MME-CoT.**

## C.2. Preliminary Categorization Result

*Table 4.* **Accuracy of MMT-Bench for different subcategories**. ACT: Action Understanding; AUT: Attribute Similarity; CNT: Cartoon Understanding; CIM: Counting; DOC: Diagram Understanding; EMO: Difference Spotting; HAL: Geographic Understanding; IIT: Image-Text Matching; IRT: Ordering; IQT: Scene Understanding; MEM: Visual Grounding; MIA: Visual Retrieval; OCR: Object Recognition; PLP: Physical Layout Prediction; RRE: Relationship Extraction; TMP: Temporal Reasoning; VCP: Visual Comprehension; VCR: Visual Coherence Reasoning; VGR: Visual Generation; VIL: Visual Identification; VPU: Visual Prediction Understanding; VRE: Visual Reasoning Evaluation.

| File Name | ACT | AUT | CNT | CIM | DOC | EMO | HAL | IIT | IRT | IQT | MEM | MIA | OCR | PLP | RRE | TMP | VCP | VCR | VGR | VIL | VPU | VRE |
|---|---|---|---|---|---|---|---|---|---|---|---|---|---|---|---|---|---|---|---|---|---|---|
| GPT4o-cot | 0.60 | 0.60 | 0.44 | 0.67 | 0.79 | 0.30 | 0.71 | 0.50 | 0.63 | 0.10 | 0.85 | 0.60 | 0.77 | 0.36 | 0.76 | 0.48 | 0.86 | 0.80 | 0.49 | 0.48 | 0.82 | 0.85 |
| GPT4-direct | 0.53 | 0.60 | 0.44 | 0.67 | 0.81 | 0.23 | 0.69 | 0.33 | 0.66 | 0.25 | 0.80 | 0.43 | 0.78 | 0.42 | 0.78 | 0.36 | 0.89 | 0.85 | 0.41 | 0.37 | 0.85 | 0.85 |
| Qwen2-VL-7B-cot | 0.53 | 0.61 | 0.34 | 0.65 | 0.77 | 0.53 | 0.74 | 0.40 | 0.31 | 0.20 | 0.78 | 0.58 | 0.60 | 0.43 | 0.69 | 0.43 | 0.85 | 0.90 | 0.54 | 0.35 | 0.79 | 0.81 |
| Qwen2-VL-7B-direct | 0.49 | 0.67 | 0.40 | 0.78 | 0.75 | 0.52 | 0.73 | 0.43 | 0.31 | 0.10 | 0.78 | 0.55 | 0.60 | 0.54 | 0.69 | 0.40 | 0.85 | 0.85 | 0.67 | 0.38 | 0.85 | 0.82 |

*Table 5.* **Accuracy of MUIRBench for different subcategories**. AU: Action Understanding; AS: Attribute Similarity; CU: Cartoon Understanding; CO: Counting; DU: Diagram Understanding; DS: Difference Spotting; GU: Geographic Understanding; ITM: Image-Text Matching; OR: Ordering; SU: Scene Understanding; VG: Visual Grounding; VR: Visual Retrieval.

| File Name | AU | AS | CU | CO | DU | DS | GU | ITM | OR | SU | VG | VR |
|---|---|---|---|---|---|---|---|---|---|---|---|---|
| GPT4o-cot | 0.48 | 0.57 | 0.55 | 0.75 | 0.82 | 0.64 | 0.59 | 0.82 | 0.38 | 0.88 | 0.56 | 0.70 |
| GPT4o-direct | 0.45 | 0.62 | 0.59 | 0.50 | 0.88 | 0.62 | 0.55 | 0.86 | 0.33 | 0.74 | 0.38 | 0.77 |
| Qwen2-VL-7B-cot | 0.38 | 0.51 | 0.42 | 0.43 | 0.43 | 0.27 | 0.21 | 0.55 | 0.13 | 0.69 | 0.37 | 0.28 |
| Qwen2-VL-7B-direct | 0.39 | 0.47 | 0.44 | 0.41 | 0.40 | 0.33 | 0.25 | 0.51 | 0.13 | 0.67 | 0.31 | 0.20 |

*Table 6.* **Accuracy of OlympiadBench for the mathematics and physics subcategories**.

| File Name | Mathematics | Physics |
|---|---|---|
| GPT4o-cot | 0.25 | 0.04 |
| GPT4o-direct | 0.07 | 0.03 |
| Qwen2-VL-7B-cot | 0.05 | 0.01 |
| Qwen2-VL-7B-direct | 0.07 | 0.01 |

# D. More Experiment Details

## D.1. More Findings

***Long CoT models may be more susceptible to distraction.*** Long CoT models may demonstrate lower relevance scores compared to other models. They frequently generate content unrelated to solving the given question, corresponding to their relatively low recall scores compared to direct evaluation, like QVQ. Although a few models with short CoT, like Mulberry and LLaVA-OV-7B, also obtain a low relevance rate, we find that it is because these models may keep repeating words when dealing with specific type of questions, resulting in irrelevant judgment. The fine-grained metric reveals that models tend to lose focus when describing images, often producing exhaustive captions regardless of their relevance to the question. From Table 3, we find that this phenomenon prevails in general scenes, space-time, and OCR tasks. This behavior can significantly slow inference by generating substantial irrelevant content. Teaching long CoT models to focus on question-critical elements represents a promising direction for future research.

## D.2. Human Agreement

we conduct additional human evaluations to verify the validity of the GPT-4o assessment from two perspectives (Chen et al., 2024a; Lin et al., 2024):

1. **Human Agreement Rate**: A binary (yes/no) human evaluation to assess agreement with the model's per-step judgments.

2. **Hallucination Detection**: We assess whether any steps identified by GPT-4o are hallucinated or contain hallucinations.

Our human agreement study covers four key metrics: Recall, Precision, Relevance rate, and Reflection quality. We randomly sample 54 predictions (9 questions from each subject) from QwenVL2-72B and QvQ-72B, totaling 216 predictions and 2,368 steps. The results are shown in Table 7.

*Table 7.* **Human Evaluation Results.**

| Metric | Agreement | Hallucination |
|---|---|---|
| Recall | 98.5% | 0% |
| Precision | 94.1% | 2.1% |
| Relevance rate | 90.8% | 0% |
| Reflection quality | 86.1% | 0% |

These results demonstrate a high correlation between GPT-4o evaluations and human judgment, indicating that GPT-4o is a reliable tool for CoT evaluation. This result also indicates that all of our analysis and conclusions are valid.

# E. Error Analysis

In this section, we analyze error patterns in the LMMs' CoT and reflection process.

We categorize the CoT errors into four types and provide the examples in Figure 9 and distribution in Figure 11:

- **Visual Perception.** The model mistakenly perceives the images and causes the error.

- **Visual Reasoning.** The model fails to grasp the key information in the images and therefore gets a wrong answer.

- **Logical Reasoning.** The model exhibits a flawed logical reasoning process such as a wrong planning of the solution.

- **Calculation.** The model makes a mistake in the calculation.

As for reflection, we define that an effective reflection should either correct previous mistakes or validate correct conclusions through new insights. We examined 200 model predictions from QVQ and identified four distinct error types that hinder productive reflection. These patterns are illustrated in Fig. 10 and their distribution is shown in Fig. 12.

The four major error types are:

- **Ineffective Reflection.** The model arrives at an incorrect conclusion and, upon reflecting, continues to make incorrect adjustments. This is the most common error type and is also witnessed most frequently.

- **Incompleteness.** The model proposes new analytical approaches but does not execute them, only stopping at the initial thought. The reflection slows down the inference process without bringing any gain.

- **Repetition.** The model restates previous content or methods without introducing new insights, leading to inefficient reasoning.

- **Interference.** The model initially reaches a correct conclusion but, through reflection, introduces errors.

Understanding and mitigating these errors is crucial for improving the reliability of LMM reflection mechanisms. The analysis provides the opportunity to focus on solving specific error types to enhance the overall reflection quality.

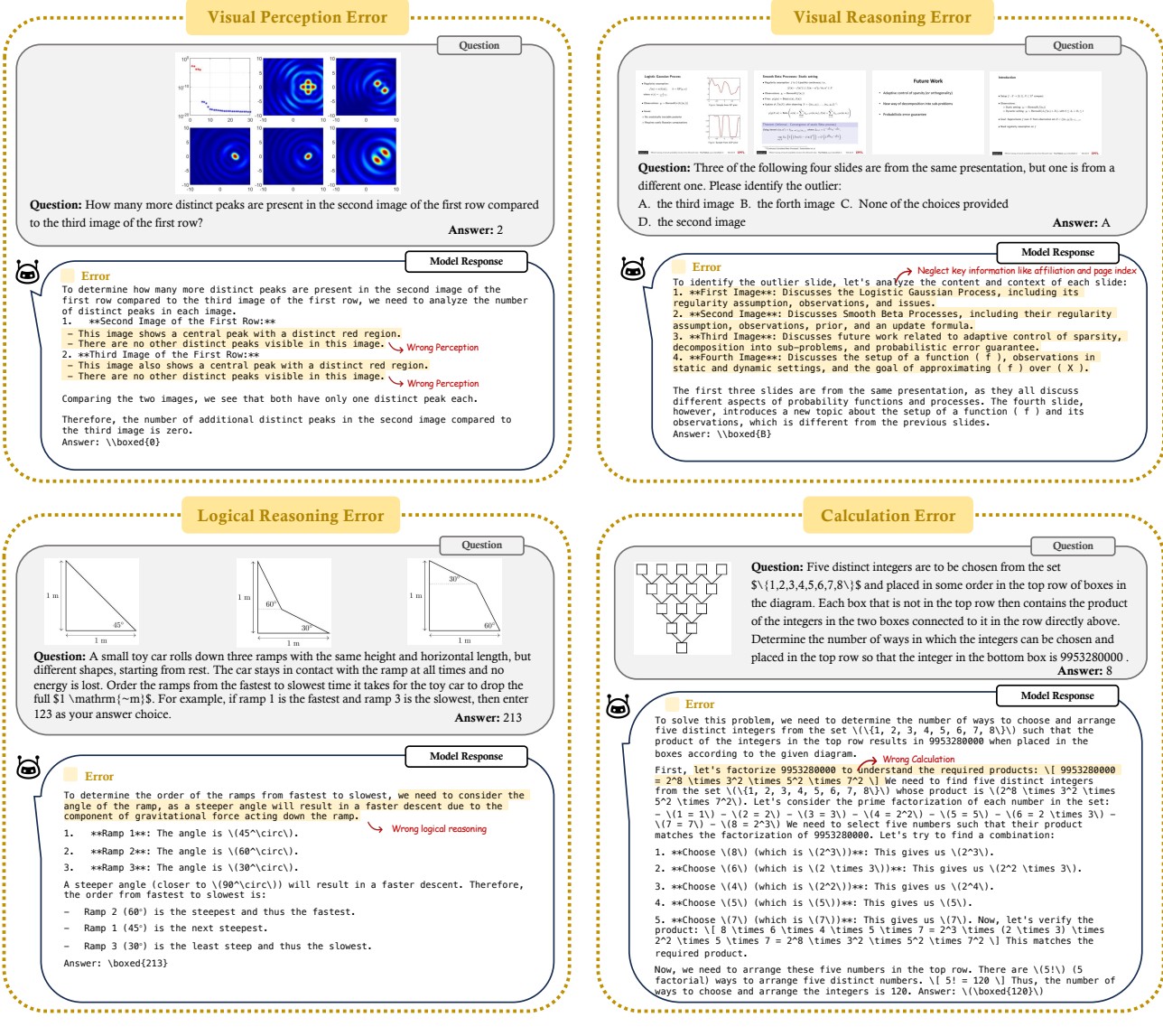

*Figure 9.* **Examples of CoT Error Types.**

## Ineffective Reflection

**Question**

**Question:** The blue car is moving forward. What is the correct order of images according to the given context?
A. the second image, the first image, the third image
B. None of the choices provided
C. the third image, the first image, the second image
D. the third image, the second image, the first image.

**Answer:** A

**Model Response**

☐ **Reflection**

↪ *Wrong Answer*

...
Alternatively, perhaps the street is not horizontal or vertical, but diagonal. Looking back at the images, the street appears to be horizontal in the aerial view, with the yellow car moving from left to right. Therefore, the sequence should be first image (left), second image (middle), third image (right). Since none of the options match this sequence, the answer must be B. none of the choices provided.

Alternatively, perhaps the question expects a different perspective. Wait, maybe the street is not horizontal but vertical. If the street is vertical, with the yellow car moving from top to bottom, then the sequence would be first image (top), third image (middle), second image (bottom).

Again, none of the options match this sequence.
...

↪ *Still Wrong Answer after Reflection*

## Repetition

**Question**

**Question:** Conversation: 2021/05/31 People1: This left picture was taken on New Year's Day. And the right picture was taken today. I've gained a lot of weight and I'm so annoyed. How many kilograms does people1 gain per month on average?
A. 21.4    B. 10.7    C. None of the choices provided    D. 1.07

**Answer:** D

**Model Response**

☐ **Reflection**

...
So, average monthly gain = total weight gain / number of months.

So, total days = 31 (January) + 28 (February) + 31 (March) + 30 (April) + 31 (May) = 151 days.
Now, to find the average monthly weight gain, perhaps I need to consider the average per month based on the number of days.

↪ *Reasoning First Time*

But that seems unnecessary. Since the weight was measured at the beginning and the end, using months as the time unit should be sufficient.

Alternatively, maybe the problem wants the average daily weight gain, and then convert it to monthly gain.
6.4 kg / 151 days = approximately 0.0424 kg per day.
0.0424 kg/day * 30 days/month = approximately 1.27 kg/month.

↪ *Repeated Reasoning*

That's roughly 1.27 kg per month, which is close to my earlier calculation of 1.28 kg per month.
...

## Incompleteness

**Question**

**Question:** Suppose there are n plates equally spaced around a circular table. Ross wishes to place an identical gift on each of k plates, so that no two neighbouring plates have gifts. Let f(n, k) represent the number of ways in which he can place the gifts. For example, f(6,3)=2, as shown below.
Determine the smallest possible value of n+k among all possible ordered pairs of integers (n, k) for which f(n, k) is a positive multiple of 2009.

**Answer:** 54

**Model Response**

☐ **Reflection**

...
Let's consider that f(n,k) is a combinatorial number, and since it's related to binomial coefficients, perhaps I can use Lucas' theorem to determine when f(n,k) is divisible by a prime p. Lucas' theorem states that C(m,p^e) = C(m_p, e_p) mod p, where m_p and e_p are the digits in base p of m and e.
However, applying Lucas' theorem directly to f(n,k) is complicated due to the circular adjacency constraints.
...

↪ *Give Up Without Trying*

## Interference

**Question**

**Question:** As shown in the figure, the straight line a parallel b, then the degree of angle 1 is ().
A:75°    B:105°    C:140°    D:145°"

**Answer:** B

**Model Response**

☐ **Reflection**

...
If angle 1 is 140°, then perhaps there's a triangle involved where the sum of angles is 180°. Let me see: if I have a triangle with angles 40°, 35°, and angle 3 (let's say), then angle 3 would be 180° − 40° − 35° = 105°.

If angle 1 is corresponding to angle 3, then it should also be 105°.

↪ *Correct Answer*
↪ *Wrong Answer*

but according to my earlier calculation, angle 1 is 140°.

Hmm, there's inconsistency here. Maybe I need to approach this differently. If angle between lines "a" and "b" is 180° − 40° = 140°, then angle 1, being corresponding to that angle, should also be 140°.
...

*Figure 10.* **Examples of Reflection Error Types.**

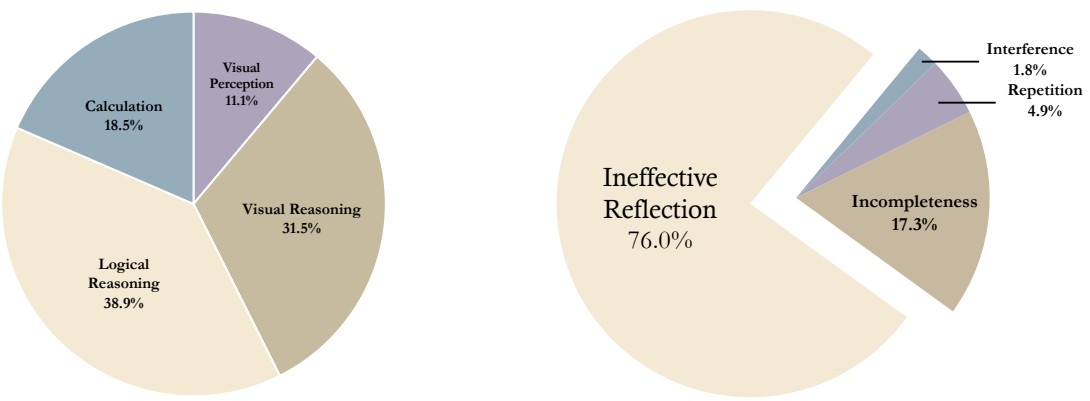

*Figure 11.* **CoT Error Distribution of MME-CoT.**

*Figure 12.* **Reflection Error Distribution of MME-CoT.**

## F. More Qualitative Examples

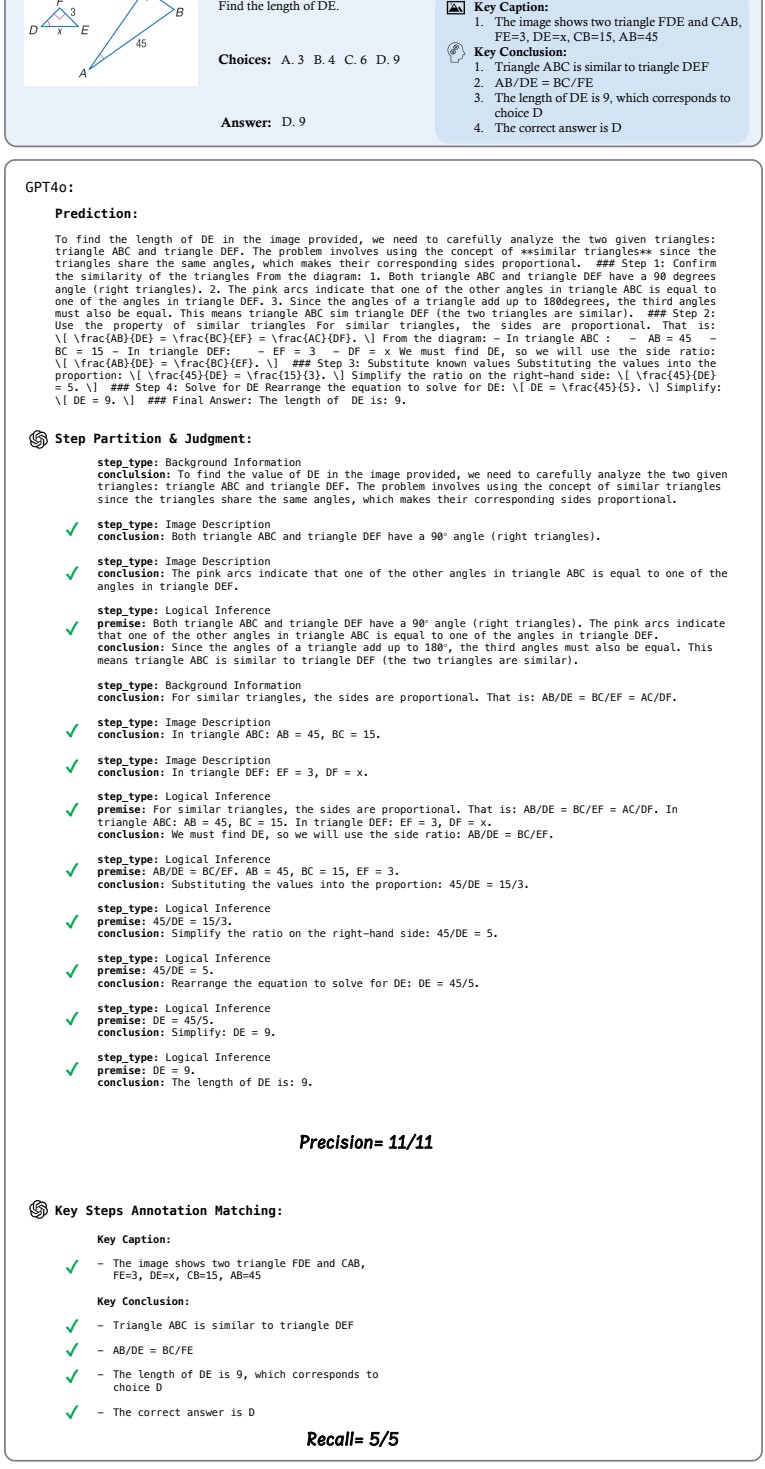

**Precision & Recall**

**Question:**
Find the length of DE.

**Choices:** A. 3  B. 4  C. 6  D. 9

**Answer:** D. 9

⭐ **Key Steps Annotation:**

🖼 **Key Caption:**
1. The image shows two triangle FDE and CAB, FE=3, DE=x, CB=15, AB=45

🧠 **Key Conclusion:**
1. Triangle ABC is similar to triangle DEF
2. AB/DE = BC/FE
3. The length of DE is 9, which corresponds to choice D
4. The correct answer is D

---

GPT4o:

**Prediction:**

To find the length of DE in the image provided, we need to carefully analyze the two given triangles: triangle ABC and triangle DEF. The problem involves using the concept of **similar triangles** since the triangles share the same angles, which makes their corresponding sides proportional. ### Step 1: Confirm the similarity of the triangles From the diagram: 1. Both triangle ABC and triangle DEF have a 90 degrees angle (right triangles). 2. The pink arcs indicate that one of the other angles in triangle ABC is equal to one of the angles in triangle DEF. 3. Since the angles of a triangle add up to 180degrees, the third angles must also be equal. This means triangle ABC sim triangle DEF (the two triangles are similar). ### Step 2: Use the property of similar triangles For similar triangles, the sides are proportional. That is: \[ \frac{AB}{DE} = \frac{BC}{EF} = \frac{AC}{DF}. \] From the diagram: – In triangle ABC : – AB = 45 – BC = 15 – In triangle DEF: – EF = 3 – DF = x We must find DE, so we will use the side ratio: \[ \frac{AB}{DE} = \frac{BC}{EF}. \] ### Step 3: Substitute known values Substituting the values into the proportion: \[ \frac{45}{DE} = \frac{15}{3}. \] Simplify the ratio on the right-hand side: \[ \frac{45}{DE} = 5. \] ### Step 4: Solve for DE Rearrange the equation to solve for DE: \[ DE = \frac{45}{5}. \] Simplify: \[ DE = 9. \] ### Final Answer: The length of  DE is: 9.

🔮 **Step Partition & Judgment:**

**step_type:** Background Information
**conclusion:** To find the value of DE in the image provided, we need to carefully analyze the two given triangles: triangle ABC and triangle DEF. The problem involves using the concept of similar triangles since the triangles share the same angles, which makes their corresponding sides proportional.

✓ **step_type:** Image Description
**conclusion:** Both triangle ABC and triangle DEF have a 90° angle (right triangles).

✓ **step_type:** Image Description
**conclusion:** The pink arcs indicate that one of the other angles in triangle ABC is equal to one of the angles in triangle DEF.

✓ **step_type:** Logical Inference
**premise:** Both triangle ABC and triangle DEF have a 90° angle (right triangles). The pink arcs indicate that one of the other angles in triangle ABC is equal to one of the angles in triangle DEF.
**conclusion:** Since the angles of a triangle add up to 180°, the third angles must also be equal. This means triangle ABC is similar to triangle DEF (the two triangles are similar).

**step_type:** Background Information
**conclusion:** For similar triangles, the sides are proportional. That is: AB/DE = BC/EF = AC/DF.

✓ **step_type:** Image Description
**conclusion:** In triangle ABC: AB = 45, BC = 15.

✓ **step_type:** Image Description
**conclusion:** In triangle DEF: EF = 3, DF = x.

✓ **step_type:** Logical Inference
**premise:** For similar triangles, the sides are proportional. That is: AB/DE = BC/EF = AC/DF. In triangle ABC: AB = 45, BC = 15. In triangle DEF: EF = 3, DF = x.
**conclusion:** We must find DE, so we will use the side ratio: AB/DE = BC/EF.

✓ **step_type:** Logical Inference
**premise:** AB/DE = BC/EF. AB = 45, BC = 15, EF = 3.
**conclusion:** Substituting the values into the proportion: 45/DE = 15/3.

✓ **step_type:** Logical Inference
**premise:** 45/DE = 15/3.
**conclusion:** Simplify the ratio on the right-hand side: 45/DE = 5.

✓ **step_type:** Logical Inference
**premise:** 45/DE = 5.
**conclusion:** Rearrange the equation to solve for DE: DE = 45/5.

✓ **step_type:** Logical Inference
**premise:** DE = 45/5.
**conclusion:** Simplify: DE = 9.

✓ **step_type:** Logical Inference
**premise:** DE = 9.
**conclusion:** The length of DE is: 9.

*Precision= 11/11*

🔮 **Key Steps Annotation Matching:**

**Key Caption:**

✓ – The image shows two triangle FDE and CAB, FE=3, DE=x, CB=15, AB=45

**Key Conclusion:**

✓ – Triangle ABC is similar to triangle DEF

✓ – AB/DE = BC/FE

✓ – The length of DE is 9, which corresponds to choice D

✓ – The correct answer is D

*Recall= 5/5*

*Figure 13.* **Examples of Precision and Recall Evaluation.**

```
Qwen2-7b:
```

**Prediction:**

To find the length of DE, we can use the property of similar triangles. Since triangles DEF and ABC are similar, the corresponding sides are proportional. We can set up the following proportion: \[ \frac{DE}{AB} = \frac{EF}{BC} \]  Given that EF = 3, BC = 15, and AB = 45, we can substitute these values into the proportion:   \[ \frac{DE}{45} = \frac{3}{15} \]    Simplifying the right side of the equation: \[ \frac{DE}{45} = \frac{1}{5} \]  To find DE, we can cross-multiply: \[ 5 \cdot DE = 45 \cdot 1 \]  \[ 5 \cdot DE = 45 \]  \[ DE = \frac{45}{5} \]  \[ DE = 9 \]  Therefore, the length of DE is 9. The correct answer is D: 9.

**Step Partition & Judgment:**

> **step_type:** background information
> **conclusion:** To find the length of DE, we can use the property of similar triangles.

> ✓ **step_type:** logical inference
> **premise:** triangles DEF and ABC are similar.
> **conclusion:** The corresponding sides are proportional.

> ✓ **step_type:** logical inference
> **premise:** the corresponding sides are proportional.
> **conclusion:** We can set up the following proportion: DE/AB = EF/BC.

> **step_type:** background information
> **conclusion:** Given that EF = 3, BC = 15, and AB = 45.

> ✓ **step_type:** logical inference
> **premise:** DE/AB = EF/BC and EF = 3, BC = 15, AB = 45.
> **conclusion:** Substitute these values into the proportion: DE/45 = 3/15.

> ✓ **step_type:** logical inference
> **premise:** DE/45 = 3/15.
> **conclusion:** Simplifying the right side of the equation: DE/45 = 1/5.

> ✓ **step_type:** logical inference
> **premise:** DE/45 = 1/5.
> **conclusion:** To find DE, we can cross-multiply: 5 * DE = 45 * 1.

> ✓ **step_type:** logical inference
> **premise:** 5 * DE = 45.
> **conclusion:** DE = 45/5.

> ✓ **step_type:** logical inference
> **premise:** DE = 45/5.
> **conclusion:** DE = 9.

> ✓ **step_type:** logical inference
> **premise:** DE = 9.
> **conclusion:** The length of DE is 9. The correct answer is D: 9.

*Precision= 8/8*

**Key Steps Annotation Matching:**

> **Key Caption:**

> ✓ − The image shows two triangle FDE and CAB,
> FE=3, DE=x, CB=15, AB=45

> **Key Conclusion:**

> ✓ − Triangle ABC is similar to triangle DEF

> ✓ − AB/DE = BC/FE

> ✓ − The length of DE is 9, which corresponds to
> choice D

> ✓ − The correct answer is D

*Recall= 5/5*

*Figure 14.* **Examples of Precision and Recall Evaluation.**

QVQ-72B-Preview:

**Prediction:**

So, I've got this geometry problem here, and I need to find the length of DE. Let's see what information is given. There are two triangles: one is ADE, and the other is ABC. From the diagram, it looks like triangle ADE is similar to triangle ABC because they have corresponding angles that are equal. Specifically, angle D in triangle ADE corresponds to angle A in triangle ABC, angle E in triangle ADE corresponds to angle B in triangle ABC, and angle F in triangle ADE corresponds to angle C in triangle ABC. Wait, actually, looking closer, triangle ADE is similar to triangle ABC, with angle D corresponding to angle A, angle E corresponding to angle B, and angle F corresponding to angle C. But in the diagram, it's labeled as points D, E, F for triangle ADE, and points A, B, C for triangle ABC. Hmm, maybe I need to be careful with the labeling. Let's look at the coordinates or the labels again. In triangle ADE, points are D, E, F, and in triangle ABC, points are A, B, C. There's a side labeled FE with length 3, and sides BC and DE are to be compared. Wait, no. In triangle ADE, sides are DE, DF, and EF, with EF being 3 units. In triangle ABC, sides are AB, BC, and AC, with BC being 15 units and AB being 45 units. There's also an angle in triangle ABC that's 90 degrees, specifically angle C is 90 degrees, since there's a red square at point C, which typically denotes a right angle. So, triangle ABC is a right-angled triangle with right angle at C, AB is 45 units, and BC is 15 units. Now, since triangle ADE is similar to triangle ABC, their corresponding sides are proportional. First, I need to identify which sides correspond to each other. Given that angle F in triangle ADE corresponds to angle C in triangle ABC, and angle C is the right angle, so angle F must also be the right angle in triangle ADE. Wait, but in triangle ADE, angles D, E, and F are all acute in the diagram, so maybe that's not the case. Actually, perhaps I misread the diagram. Let's assume that triangle ADE is similar to triangle ABC, with angle at A being common to both. But in the diagram, it seems like triangle ADE is positioned with point A at the origin, and triangle ABC is another triangle with point A shared. Wait, perhaps point A is the common vertex. Let me try to rephrase the problem based on the diagram description. Points D, E, F form a triangle with EF = 3 units. Points A, B, C form another triangle with BC = 15 units and AB = 45 units, and angle at C is 90 degrees. There's a line from A to C, and another line from A to E. Wait, maybe I need to consider the similarity ratio between the two triangles. Since triangle ADE is similar to triangle ABC, the ratio of their corresponding sides should be equal. Let's denote the lengths: Let DE = x (which is what we need to find) EF = 3 units In triangle ABC, BC = 15 units, AB = 45 units, and angle C = 90 degrees. Wait, but in triangle ADE, which sides correspond to which sides in triangle ABC? If angle at D corresponds to angle at A, angle at E corresponds to angle at B, and angle at F corresponds to angle at C, then: DE corresponds to AB, DF corresponds to AC, and EF corresponds to BC. Therefore, the ratio of corresponding sides is DE / AB = DF / AC = EF / BC. But we don't know the length of DF or AC. Wait, perhaps I need to find another way. Since triangle ABC is a right-angled triangle with right angle at C, and sides BC = 15 and AB = 45, I can find AC using the Pythagorean theorem. In triangle ABC: AB² = AC² + BC² 45² = AC² + 15² 2025 = AC² + 225 AC² = 2025 − 225 = 1800 AC = sqrt(1800) = sqrt(100∗18) = 10∗sqrt(18) = 10∗3∗sqrt(2) = 30∗sqrt(2) So, AC = 30∗sqrt(2) Now, since triangle ADE is similar to triangle ABC, with DE corresponding to AB, DF corresponding to AC, and EF corresponding to BC. Therefore, the ratio DE / AB = DF / AC = EF / BC We can write: x / 45 = DF / (30∗sqrt(2)) = 3 / 15 Simplify 3 / 15: that's 1 / 5 So, x / 45 = 1 / 5 Therefore, x = 45 ∗ (1 / 5) = 9 So, DE = 9 units Therefore, the correct choice is D:9 ∗∗Final Answer∗∗: D.

⊛ **Step Partition & Judgment:**

    **step_type:** background information
    **conclusion:** there are two triangles: one is ADE, and the other is ABC.

✓  **step_type:** image description
    **conclusion:** triangle ADE is similar to triangle ABC because they have corresponding angles that are equal.

✓  **step_type:** image description
    **conclusion:** in triangle ADE, points are D, E, F, and in triangle ABC, points are A, B, C.

✓  **step_type:** image description
    **conclusion:** there's a side labeled FE with length 3.

✓  **step_type:** image description
    **conclusion:** in triangle ABC, sides are AB, BC, and AC, with BC being 15 units and AB being 45 units.

✓  **step_type:** image description
    **conclusion:** there's an angle in triangle ABC that's 90 degrees, specifically angle C is 90 degrees.

✓  **step_type:** logical inference
    **premise:** triangle ABC is a right-angled triangle with right angle at C, AB is 45 units, and BC is 15 units.
    **conclusion:** since triangle ADE is similar to triangle ABC, their corresponding sides are proportional.

✗  **step_type:** logical inference
    **premise:** given that angle F in triangle ADE corresponds to angle C in triangle ABC, and angle C is the right angle.
    **conclusion:** angle F must also be the right angle in triangle ADE.

✓  **step_type:** logical inference
    **premise:** triangle ADE is similar to triangle ABC, with angle at A being common to both.
    **conclusion:** points D, E, F form a triangle with EF = 3 units.

✓  **step_type:** logical inference
    **premise:** points A, B, C form another triangle with BC = 15 units and AB = 45 units, and angle at C is 90 degrees.
    **conclusion:** there's a line from A to C, and another line from A to E.

✓  **step_type:** logical inference
    **premise:** since triangle ADE is similar to triangle ABC, the ratio of their corresponding sides should be equal.
    **conclusion:** let DE = x (which is what we need to find), EF = 3 units.

✓  **step_type:** logical inference
    **premise:** in triangle ABC, BC = 15 units, AB = 45 units, and angle C = 90 degrees.
    **conclusion:** the ratio of corresponding sides is DE / AB = DF / AC = EF / BC.

✓  **step_type:** logical inference
    **premise:** we don't know the length of DF or AC.
    **conclusion:** perhaps I need to find another way.

✓  **step_type:** logical inference
    **premise:** triangle ABC is a right-angled triangle with right angle at C, and sides BC = 15 and AB = 45.
    **conclusion:** I can find AC using the Pythagorean theorem.

✓  **step_type:** logical inference
    **premise:** in triangle ABC: AB² = AC² + BC², 45² = AC² + 15².
    **conclusion:** AC² = 2025 − 225 = 1800, AC = sqrt(1800) = 30∗sqrt(2).

✓  **step_type:** logical inference
    **premise:** since triangle ADE is similar to triangle ABC, with DE corresponding to AB, DF corresponding to AC, and EF corresponding to BC.
    **conclusion:** the ratio DE / AB = DF / AC = EF / BC.

✓  **step_type:** logical inference
    **premise:** x / 45 = DF / (30∗sqrt(2)) = 3 / 15.
    **conclusion:** simplify 3 / 15: that's 1 / 5.

✓  **step_type:** logical inference
    **premise:** x / 45 = 1 / 5.
    **conclusion:** x = 45 ∗ (1 / 5) = 9.

✓  **step_type:** logical inference
    **premise:** x = 9.
    **conclusion:** DE = 9 units.

✓  **step_type:** logical inference
    **premise:** DE = 9 units.
    **conclusion:** the correct choice is D:9.

*Precision= 18/19*

*Figure 15.* **Examples of Precision and Recall Evaluation.**

```
QVQ-72B-Preview:
```

🅖 **Annotation Steps Matching:**

      **Key Caption:**

✓     – The image shows two triangle FDE and CAB,
        FE=3, DE=x, CB=15, AB=45

      **Key Conclusion:**

✗     – Triangle ABC is similar to triangle DEF

✓     – AB/DE = BC/FE

✓     – The length of DE is 9, which corresponds to
        choice D

✓     – The correct answer is D

*Recall= 4/5*

*Figure 16.* **Examples of Precision and Recall Evaluation.**

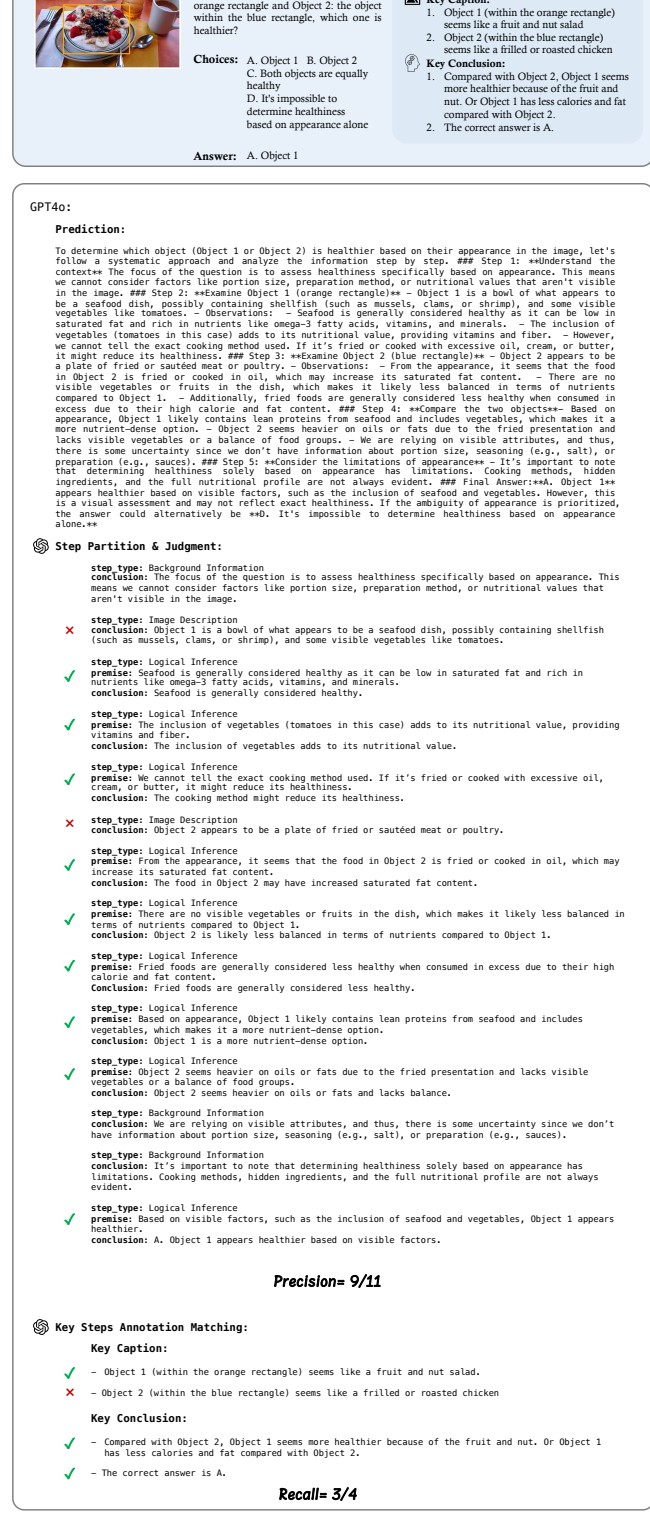

*Figure 17.* **Examples of Precision and Recall Evaluation.**

```
Qwen2-7b:
```

**Prediction:**

To determine which object is healthier between Object 1 (the object within the orange rectangle) and Object 2 (the object within the blue rectangle), let's analyze each object based on common health criteria: 1. **Object 1 (the object within the orange rectangle)**: – This appears to be a bowl of yogurt topped with various fruits, nuts, and possibly granola. – Yogurt is a good source of protein and probiotics. – Fruits provide vitamins, minerals, and fiber. – Nuts and granola add healthy fats and fiber. – Overall, this looks like a balanced and nutritious meal. 2. **Object 2 (the object within the blue rectangle)**: – This appears to be a plate of fried food, possibly pancakes or waffles. – Fried foods are typically high in calories, saturated fats, and sodium. – They may also be low in fiber and other essential nutrients. – This does not appear to be a healthy choice compared to the yogurt bowl. Based on this analysis, **Object 1 (the yogurt bowl with fruits and granola)** is healthier than **Object 2 (the fried food)**. **Final Answer: A. Object 1**

**Step Partition & Judgment:**

**step_type:** background information
**conclusion:** To determine which object is healthier between Object 1 (the object within the orange rectangle) and Object 2 (the object within the blue rectangle), let's analyze each object based on common health criteria.

✓ **step_type:** image description
**conclusion:** Object 1 (the object within the orange rectangle) appears to be a bowl of yogurt topped with various fruits, nuts, and possibly granola.

**step_type:** background information
**conclusion:** Yogurt is a good source of protein and probiotics.

**step_type:** background information
**conclusion:** Fruits provide vitamins, minerals, and fiber.

**step_type:** background information
**conclusion:** Nuts and granola add healthy fats and fiber.

✓ **step_type:** logical inference
**premise:** yogurt is a good source of protein and probiotics. Fruits provide vitamins, minerals, and fiber. Nuts and granola add healthy fats and fiber.
**conclusion:** Overall, this looks like a balanced and nutritious meal.

✗ **step_type:** image description
**conclusion:** Object 2 (the object within the blue rectangle) appears to be a plate of fried food, possibly pancakes or waffles.

**step_type:** background information
**conclusion:** Fried foods are typically high in calories, saturated fats, and sodium.

**step_type:** background information
**conclusion:** They may also be low in fiber and other essential nutrients.

✓ **step_type:** logical inference
**premise:** fried foods are typically high in calories, saturated fats, and sodium. They may also be low in fiber and other essential nutrients.
**conclusion:** This does not appear to be a healthy choice compared to the yogurt bowl.

✓ **step_type:** logical inference
**premise:** overall, this looks like a balanced and nutritious meal. This does not appear to be a healthy choice compared to the yogurt bowl.
**conclusion:** Based on this analysis, Object 1 (the yogurt bowl with fruits and granola) is healthier than Object 2 (the fried food).

✓ **step_type:** logical inference
**premise:** based on this analysis, Object 1 (the yogurt bowl with fruits and granola) is healthier than Object 2 (the fried food).
**conclusion:** Final Answer: A. Object 1

*Precision= 5/6*

**Key Steps Annotation Matching:**

**Key Caption:**

✓ – Object 1 (within the orange rectangle) seems like a fruit and nut salad.

✗ – Object 2 (within the blue rectangle) seems like a frilled or roasted chicken

**Key Conclusion:**

✓ – Compared with Object 2, Object 1 seems more healthier because of the fruit and nut. Or Object 1 has less calories and fat compared with Object 2.

✓ – The correct answer is A.

*Recall= 3/4*

*Figure 18.* **Examples of Precision and Recall Evaluation.**

```
QVQ-72B-Preview:
```

**Prediction:**

Alright, I've got this question here. There are two objects, each within a colored rectangle: one in orange and one in blue. The task is to determine which one is healthier. The options are A. Object 1 (orange rectangle), B. Object 2 (blue rectangle), C. Both are equally healthy, or D. It's impossible to determine based on appearance alone.  First, I need to understand what's in each rectangle.  Object 1 is in the orange rectangle. From the description, it sounds like a breakfast setup. There's a plate with what appears to be yogurt topped with fruits like bananas, strawberries, and possibly blueberries, along with some granola. There's also a glass of iced tea, a mug of coffee, and some roasted potatoes in the background. Object 2 is in the blue rectangle. It's a plate of roasted potatoes.  So, the comparison is between a fruit and granola yogurt bowl and a plate of roasted potatoes.  Now, I need to assess which of these is healthier. Let's consider the nutrients each provides.  The yogurt bowl with fruits and granola:  – Yogurt is a good source of protein and calcium.  – Bananas provide potassium and vitamin C.  – Strawberries and blueberries are rich in antioxidants and vitamin C.  – Granola can vary, but it often contains oats, which are high in fiber, and possibly nuts or seeds for healthy fats and additional protein.  The roasted potatoes:  – Potatoes are a source of carbohydrates, vitamin C, and potassium.  – How they're roasted matters: if it's with minimal oil and seasoning, they can be quite healthy. However, if they're deeply fried or cooked with a lot of fat, they might be higher in unhealthy fats.  Given that, the yogurt bowl seems to offer a more balanced meal with a combination of protein, carbohydrates, fiber, and vitamins from the fruits and granola. The roasted potatoes, while they can be part of a healthy diet, especially if prepared healthily, don't provide as diverse a range of nutrients as the yogurt bowl.  Moreover, depending on the preparation, the potatoes might be higher in calories from fat compared to the yogurt bowl.  But, to be fair, if the potatoes are roasted with very little oil, they could still be quite healthy.  However, in general, the yogurt bowl with fruits and granola is likely to be a healthier choice due to its nutrient density and diversity.  Therefore, my answer would be A. Object 1.  But I should consider if there's any way that the potatoes could be healthier in some scenarios.  For example, if the granola is high in sugar or if the portion sizes are not considered, the potatoes might be a better option in those specific cases.  But, based on the visual information provided, the yogurt bowl looks well-balanced and nutritious.  So, I think A. Object 1 is the correct answer.  **Final Answer** A.

ⓢ **Step Partition & Judgment:**

    **step_type:** background information
    **conclusion:** There are two objects, each within a colored rectangle: one in orange and one in blue. The task is to determine which one is healthier. The options are A. Object 1 (orange rectangle), B. Object 2 (blue rectangle), C. Both are equally healthy, or D. It's impossible to determine based on appearance alone.

✓    **step_type:** image description
    **conclusion:** Object 1 is in the orange rectangle. There's a plate with what appears to be yogurt topped with fruits like bananas, strawberries, and possibly blueberries, along with some granola. There's also a glass of iced tea, a mug of coffee, and some roasted potatoes in the background.

✓    **step_type:** image description
    **conclusion:** Object 2 is in the blue rectangle. It's a plate of roasted potatoes.

✓    **step_type:** logical inference
    **premise:** comparison between a fruit and granola yogurt bowl and a plate of roasted potatoes.
    **conclusion:** The yogurt bowl seems to offer a more balanced meal with a combination of protein, carbohydrates, fiber, and vitamins from the fruits and granola.

✓    **step_type:** logical inference
    **premise:** the roasted potatoes, while they can be part of a healthy diet, especially if prepared healthily, don't provide as diverse a range of nutrients as the yogurt bowl.
    **conclusion:** The yogurt bowl is likely to be a healthier choice due to its nutrient density and diversity.

✓    **step_type:** logical inference
    **premise:** based on the visual information provided, the yogurt bowl looks well-balanced and nutritious.
    **conclusion:** A. Object 1 is the correct answer.

*precision: 5/5*

ⓢ **Key Steps Annotation Matching:**

    **Key Caption:**

✓    – Object 1 (within the orange rectangle) seems like a fruit and nut salad.

✗    – Object 2 (within the blue rectangle) seems like a frilled or roasted chicken

    **Key Conclusion:**

✓    – Compared with Object 2, Object 1 seems more healthier because of the fruit and nut. Or Object 1 has less calories and fat compared with Object 2.

✓    – The correct answer is A.

*recall: 3/4*

*Figure 19.* **Examples of Precision and Recall Evaluation.**

# Relevance Rate

**Question:**

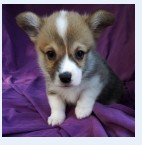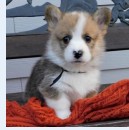

How many puppies in the images are sticking out their tongues?

**Choices:** A. Zero  B. Two  C. Three
D. None of the choices provided
E. Four

**Answer:** A. Zero

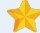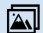 **Key Steps Annotation:**

**Key Caption:**
1. In the first image, there is one puppy, while its tongue is not sticking out.
2. In the second image, there is one puppy, while its tongue is not sticking out.

**Key Conclusion:**
1. no puppies are sticking out their tongues.
2. The answer is A.

---

**GPT4o:**

`............ irrelevant information ............`

**Prediction:**

Let's carefully analyze the images step by step: ### 1. **Image 1 Analysis:** – The puppy is sitting on an orange blanket. – The position of the mouth and tongue of the puppy is visible. – Upon closer inspection, **the puppy is not sticking out its tongue.** ### 2. **Image 2 Analysis:** – The puppy is sitting on a purple background. – The position of the mouth and tongue is also visible here. – Upon careful observation, **the puppy is again not sticking out its tongue.** ### Intermediate Observations: – In both images, the puppies are not sticking out their tongues. ### Final Answer: **A. Zero**

**Judgments:**

✗ **step_type:** image description
**conclusion:** The puppy is sitting on an orange blanket.

✓ **step_type:** image description
**conclusion:** The position of the mouth and tongue of the puppy is visible.

✓ **step_type:** logical inference
**conclusion:** The puppy is not sticking out its tongue.

✗ **step_type:** image description
**conclusion:** The puppy is sitting on a purple background.

✓ **step_type:** image description
**conclusion:** The position of the mouth and tongue is also visible here.

✓ **step_type:** logical inference
**conclusion:** The puppy is again not sticking out its tongue.

✓ **step_type:** logical inference
**conclusion:** In both images, the puppies are not sticking out their tongues.

✓ **step_type:** logical inference
**conclusion:** Final answer is A. Zero.

### Raw Relevance Rate = 6/8

---

*Figure 20.* **Examples of Relevance Rate Evaluation.**

```
Qwen2-7b:
```

⌐ irrelevant information ⌐

**Prediction:**

To determine how many puppies are sticking out their tongues in each image, let's analyze each image step by step: 1. **First Image**:    – There is one puppy in the image.    – The puppy is not sticking out its tongue. 2. **Second Image**:    – There is one puppy in the image.    – The puppy is not sticking out its tongue.  Since neither puppy in either image is sticking out its tongue, the correct answer is:  A. Zero

**Judgments:**

✓ **step_type:** background information
**conclusion:** to determine how many puppies are sticking out their tongues in each image, let's analyze each image step by step.

✓ **step_type:** image description
**conclusion:** there is one puppy in the first image.

✓ **step_type:** image description
**conclusion:** the puppy in the first image is not sticking out its tongue.

✓ **step_type:** image description
**conclusion:** there is one puppy in the second image.

✓ **step_type:** image description
**conclusion:** the puppy in the second image is not sticking out its tongue.

✓ **step_type:** logical inference
**conclusion:** since neither puppy in either image is sticking out its tongue, the correct answer is: A. Zero.

### Raw Relevance Rate = 6/6

*Figure 21.* **Examples of Relevance Rate Evaluation.**

```
QVQ-72B-Preview:
```

**Prediction:**

Alright, I've got this question here. It says, "How many puppies are sticking out their tongues in each image?" And there are two images provided. Okay, so I need to look at both images and count how many puppies are sticking out their tongues in each one. First, let's look at the first image. There's a corgi puppy sitting on an orange blanket against a wooden background with some decorative metalwork. The puppy has a tri-color coat—white, black, and tan—and it's wearing a black collar. Its tongue is not sticking out. So, in this image, the number of puppies with their tongues out is zero. Now, moving on to the second image. This one features another corgi puppy, this time against a purple fabric background. This puppy also has a tri-color coat and is looking directly at the camera. I need to check if its tongue is sticking out. Hmm, from what I can see, the puppy's mouth is closed, and there's no tongue visible. So, again, zero puppies with their tongues out in this image. Wait a minute, the question says "each image," and there are two images. So, for each image, I need to count the number of puppies sticking out their tongues. In both images, there's only one puppy, and neither of them is sticking out their tongue. So, for both images, the count is zero. But let's double-check. Maybe there's something I'm missing. In the first image, the puppy is clearly not showing its tongue. In the second image, similarly, no tongue is visible. So, I think it's safe to say that in each image, there are zero puppies sticking out their tongues. Looking back at the options provided: A . Zero, B. Two, C. Three, D. None of the choices provided, E. Four. Since in each image, the number is zero, and there are two images, but the question specifies "in each image," so the answer should be A. Zero. Wait, but maybe there's a trick here. Maybe "each image" refers to all images combined. But no, the question specifically says "in each image," so I think it's asking for the number per image, not in total across all images. So, my final answer is A. Zero. Final Answer: A. 0

**Judgments:**

✓  **step_type:** background information
   **conclusion:** The question asks how many puppies are sticking out their tongues in each image.

✗  **step_type:** image description
   **conclusion:** The first image shows a corgi puppy on an orange blanket with a tri-color coat and a black collar.

✓  **step_type:** logical inference
   **conclusion:** In the first image, the puppy's tongue is not sticking out, so the count is zero.

✗  **step_type:** image description
   **conclusion:** The second image shows a corgi puppy against a purple fabric background with a tri-color coat.

✓  **step_type:** logical inference
   **conclusion:** In the second image, the puppy's mouth is closed, so the count is zero.

✓  **step_type:** logical inference
   **conclusion:** For each image, the number of puppies sticking out their tongues is zero.

✓  **step_type:** logical inference
   **conclusion:** Double-checking confirms zero puppies sticking out their tongues in both images.

✓  **step_type:** background information
   **conclusion:** Reviewing the answer choices: A. Zero, B. Two, C. Three, D. None of the choices provided, E. Four.

✓  **step_type:** logical inference
   **conclusion:** The answer is A. Zero, as the count is zero in each image.

✓  **step_type:** logical inference
   **conclusion:** Considering if 'each image' refers to all images combined, but it specifies per image.

✓  **step_type:** logical inference
   **conclusion:** Final answer is A. Zero.

*Raw Relevance Rate = 9/11*

*Figure 22.* **Examples of Relevance Rate Evaluation.**

# Reflection Quality

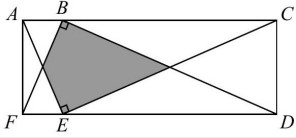

**Question:**

In the diagram, ACDF is a rectangle with AC=200 and CD=50. Also, triangle FBD and triangle AEC are congruent triangles which are right-angled at B and E, respectively. What is the area of the shaded region?

**Answer:** 2500

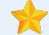 **Key Steps Annotation:**

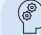 **Key Conclusion:**
1. join BE
2. BAFE is a rectangle
3. BCDE is a rectangle
4. 1/4 of rectangle ABEF is shaded and 1/4 of rectangle BCDE is shaded
5. The total shaded area is 1/4(10000) = 2500
6. The answer is 2500

---

**QVQ-72B-Preview:**

`redundant reflection`

**Prediction:**

…
Now, triangle FBD has a right angle at B, so B is somewhere on FD, I think. Wait, no, FD is the bottom side of the rectangle. But if FBD is a triangle with vertices at F, B, and D, and it's right-angled at B, then B must be one of the endpoints, but that doesn't make sense. Wait, maybe B is not on FD. Similarly, triangle AEC is right-angled at E, and E is on FD, perhaps. Wait, maybe I need to think differently. Since AC = 200 and CD = 50, and the triangles are congruent right-angled triangles, perhaps the legs of these triangles are equal in length. Let's denote the legs of the right-angled triangles. Let's say, for triangle FBD, FB and BD are the legs, and for triangle AEC, AE and EC are the legs. Given that the triangles are congruent, FB = EC and BD = AE. But since both triangles are right-angled and congruent, their corresponding sides must be equal. Wait, but I need to find the area of the shaded region, which is the overlapping area of these two triangles. Maybe I should find the area of one triangle and then double it, and then subtract the non-overlapping parts. Alternatively, perhaps I can find the area of the intersection directly. Let me try to find the equations of the lines that form the triangles to find their intersection points.
…

**Judgments:**

✓ **conclusion:** B is not on FD, reconsidering the position of B

✗ **conclusion:** The legs of the triangles are equal, reconsidering the length of the sides

✗ **conclusion:** The goal is to find the area of the shaded region, reconsidering the method of calculating it
…

*Figure 23.* **Examples of Reflection Quality Evaluation.**

# G. Detailed Evaluation Setup

## G.1. CoT Quality Evaluation Prompts

**Recall Evaluation Prompt**

You are an expert system to verify solutions to image-based problems. Your task is to match the ground truth middle steps with the provided solution.

INPUT FORMAT:
1. Problem: The original question/task
2. A Solution of a model
3. Ground Truth: Essential steps required for a correct answer

MATCHING PROCESS:

You need to match each ground truth middle step with the solution:

Match Criteria:
- The middle step should exactly match in the content or is directly entailed by a certain content in the solution
- All the details must be matched, including the specific value and content
- You should judge all the middle steps for whether there is a match in the solution

OUTPUT FORMAT:
```
[
  {
    "step_index": \textless integer\textgreater,
    "judgment": "Matched" | "Unmatched"
  }
]
```
ADDITIONAL RULES:
1. Only output the JSON array with no additional information.
2. Judge each ground truth middle step in order without omitting any step.

Here are the problem, answer, solution, and ground truth middle steps:

[Problem]

{question}

[Answer]

{answer}

[Solution]

{solution}

[Ground Truth Information]

{gt_annotation}

Precision Evaluation Prompt

# Task Overview
Given a solution with multiple reasoning steps for an image-based problem, reformat it into well-structured steps and evaluate their correctness.

# Step 1: Reformatting the Solution
Convert the unstructured solution into distinct reasoning steps while:
- Preserving all original content and order
- Not adding new interpretations
- Not omitting any steps

## Step Types
1. Logical Inference Steps
- Contains exactly one logical deduction
- Must produce a new derived conclusion
- Cannot be just a summary or observation

2. Image Observation Steps
- Pure visual observations
- Only includes directly visible elements
- No inferences or assumptions

3. Background Information Steps
- External knowledge or question context
- No inference process involved

## Step Requirements
- Each step must be atomic (one conclusion per step)
- No content duplication across steps
- Initial analysis counts as background information
- Final answer determination counts as logical inference

# Step 2: Evaluating Correctness
Evaluate each step against:

## Ground Truth Matching
For image observations:
- Key elements must match ground truth observations

For logical inferences:
- Conclusion must EXACTLY match or be DIRECTLY entailed by ground truth

## Reasonableness Check (if no direct match)
Step must:
- Premises must not contradict any ground truth or correct answer
- Logic is valid
- Conclusion must not contradict any ground truth
- Conclusion must support or be neutral to correct answer

## Judgement Categories
- "Match": Aligns with ground truth
- "Reasonable": Valid but not in ground truth

- "Wrong": Invalid or contradictory
- "N/A": For background information steps

# Output Requirements
1. The output format must be in valid JSON format without any other content.
2. For highly repetitive patterns, output it as a single step.
3. Output maximum 40 steps. Always include the final step that contains the answer.

Here is the json output format:
## Output Format

```
[
  {
    "step_type": "image observation|logical inference|background information",
    "premise": "Evidence (only for logical inference)",
    "conclusion": "Step result",
    "judgment": "Match|Reasonable|Wrong|N/A"
  }
]
```

Here is the problem, and the solution that needs to be reformatted to steps:

[Problem]

{question}

[Solution]

{solution}

[Correct Answer]

{answer}

[Ground Truth Information]

{gt_annotation}

## G.2. CoT Efficiency Prompt

Relevance Rate Evaluation Prompt

# Task Overview Given a solution with multiple reasoning steps for an image-based problem, evaluate the relevance to get a solution (ignore correct or wrong) of each step.

# Step 1: Reformatting the Solution Convert the unstructured solution into distinct reasoning steps while:
- Preserving all original content and order
- Not adding new interpretations
- Not omitting any steps

## Step Types
1. Logical Inference Steps
- Contains exactly one logical deduction

- Must produce a new derived conclusion
- Cannot be just a summary or observation
2. Image Description Steps
- Pure visual observations
- Only includes directly visible elements
- No inferences or assumptions
3. Background Information Steps
- External knowledge or question context
- No inference process involved

## Step Requirements - Each step must be atomic (one conclusion per step)
- No content duplication across steps
- Initial analysis counts as background information
- Final answer determination counts as logical inference

# Step 2: Evaluating Relevancy
A relevant step is considered as: 75% content of the step must be related to trying to get a solution (ignore correct or wrong) to the question.

IMPORTANT NOTE:
Evaluate relevancy independent of correctness. As long as the step is trying to get to a solution, it is considered relevant. Logical fallacy, knowledge mistake, inconsistent with previous steps, or other mistakes do not affect relevance. A logically wrong step can be relevant if the reasoning attempts to address the question.

The following behaviour is considered as relevant:
i. The step is planning, summarizing, thinking, verifying, calculating, or confirming an intermediate/final conclusion helpful to get a solution.
ii. The step is summarizing or reflecting on previously reached conclusion relevant to get a solution.
iii. Repeating the information in the question or give the final answer.
iv. A relevant image depiction should be in one of following situation:
1. help to obtain a conclusion helpful to solve the question later;
2. help to identify certain patterns in the image later;
3. directly contributes to the answer
v. Depicting or analyzing the options of the question is also relevant.
vi. Repeating previous relevant steps are also considered relevant.

The following behaviour is considered as irrelevant:
i. Depicting image information that does not related to what is asking in the question. Example: The question asks how many cars are present in all the images. If the step focuses on other visual elements like the road or building, the step is considered as irrelevant.
ii. Self-thought not related to what the question is asking.
iii. Other information that is tangential for answering the question.

# Output Format

```
[
  {
    "step_type": "image observation|logical inference|background information",
    "conclusion": "A brief summary of step result",
    "relevant": "Yes|No"
  }
]
```

# Output Rules

Direct JSON output without any other output
Output at most 40 steps

Here is the problem, and the solution that needs to be reformatted to steps:
[Problem]

{question}

[Solution]

{solution}

---

## Reflection Quality Evaluation Prompt

Here's a refined prompt that improves clarity and structure:

# Task
Evaluate reflection steps in image-based problem solutions, where reflections are self-corrections or reconsideration of previous statements.

# Reflection Step Identification
Reflections typically begin with phrases like:
- "But xxx"
- "Alternatively, xxx"
- "Maybe I should"
- "Let me double-check"
- "Wait xxx"
- "Perhaps xxx"
It will throw a doubt of its previously reached conclusion or raise a new thought.

# Evaluation Criteria
Correct reflections must:
1. Reach accurate conclusions aligned with ground truth
2. Use new insights to find the mistake of the previous conclusion or verify its correctness.

Invalid reflections include:
1. Repetition - Restating previous content or method without new insights
2. Wrong Conclusion - Reaching incorrect conclusions vs ground truth
3. Incompleteness - Proposing but not executing new analysis methods
4. Other - Additional error types

# Input Format

[Problem]

{question}

[Solution]

{solution}

[Ground Truth]

{gt_annotation}

# Output Requirements
1. The output format must be in valid JSON format without any other content.
2. Output maximum 30 reflection steps.

Here is the json output format:
## Output Format

```
[
  {
    "conclusion": "One-sentence summary of reflection outcome",
    "judgment": "Correct|Wrong",
    "error_type": "N/A|Repetition|Wrong Conclusion|Incompleteness|Other"
  }
]
```

# Rules
1. Preserve original content and order
2. No new interpretations
3. Include ALL reflection steps
4. Empty list if no reflections found
5. Direct JSON output without any other output

## G.3. Direct Evaluation Prompt

**Answer Extraction Prompt**

You are an AI assistant who will help me to extract an answer of a question. You are provided with a question and a response, and you need to find the final answer of the question.

Extract Rule:
[Multiple choice question]
1. The answer could be answering the option letter or the value. You should directly output the choice letter of the answer.
2. You should output a single uppercase character in A, B, C, D, E, F, G, H, I (if they are valid options), and Z.
3. If the meaning of all options are significantly different from the final answer, output Z.

[Non Multiple choice question]
1. Output the final value of the answer. It could be hidden inside the last step of calculation or inference. Pay attention to what the question is asking for to extract the value of the answer.
2. The final answer could also be a short phrase or sentence.
3. If the response doesn't give a final answer, output Z.

Output Format: Directly output the extracted answer of the response.

{In Context Examples}

Question: {question}
Answer: {response}

Your output:

Answer Scoring Prompt

You are an AI assistant who will help me to judge whether two answers are consistent.

Input Illustration: [Standard Answer] is the standard answer to the question. [Model Answer] is the answer extracted from a model's output to this question.
Task Illustration: Determine whether [Standard Answer] and [Model Answer] are consistent.

Consistent Criteria:
[Multiple-Choice questions]
1. If the [Model Answer] is the option letter, then it must completely matches the [Standard Answer].
2. If the [Model Answer] is not an option letter, then the [Model Answer] must completely match the option content of [Standard Answer].
[Nan-Multiple-Choice questions]
1. The [Model Answer] and [Standard Answer] should exactly match.
2. If the meaning is expressed in the same way, it is also considered consistent, for example, 0.5m and 50cm.

Output Format: 1. If they are consistent, output 1; if they are different, output 0.
2. DIRECTLY output 1 or 0 without any other content.
{In Context Examples}

Question: {question}
[Model Answer]: {extract_answer}
[Standard Answer]: {gt_answer}
Your output:

