# OpenReview forum: "MME-CoT: Benchmarking Chain-of-Thought in Large Multimodal Models for Reasoning Quality, Robustness, and Efficiency"
_ICML.cc/2025/Conference — ICML 2025 poster_

### Official Review · Reviewer_3D9g · 2025-03-12

**Overall Recommendation:** 3

**Summary:**

This paper introduces MMECoT, a benchmark designed to evaluate the Chain-of-Thought (CoT) reasoning performance of Large Multimodal Models (LMMs) across six domains: math, science, OCR, logic, space-time, and general scenes. It proposes a comprehensive evaluation suite with three novel metrics to assess reasoning quality, robustness, and efficiency at a fine-grained level. The study analyzes state-of-the-art LMMs and uncovers key findings: 1) Models with a reflection mechanism, like QVQ, show superior CoT quality, approaching GPT-4o; 2) CoT prompting tends to degrade LMM performance on perception-heavy tasks, possibly due to overthinking; and 3) Despite high CoT quality, LMMs with reflection mechanisms are inefficient during both response and self-correction phases.

**Claims And Evidence:**

Yes, the claims are well-supported and make sense in light of the experimental results.

**Essential References Not Discussed:**

The references discussed in the paper are sufficient for understanding the context and key contributions. The study thoroughly covers the relevant advancements in Chain-of-Thought (CoT) reasoning and Large Multimodal Models (LMMs).

**Experimental Designs Or Analyses:**

Yes, the experimental designs and analyses appear to be sound based on the reported correlation studies.

**Methods And Evaluation Criteria:**

Yes, the proposed CoT benchmark for LMMs is highly meaningful, as it addresses the need for systematic evaluation of reasoning across multiple domains.

**Other Comments Or Suggestions:**

Please see weakness and suggestions above.

**Other Strengths And Weaknesses:**

**Strengths.**

This paper's strengths lie in its comprehensive approach to evaluating Chain-of-Thought (CoT) reasoning in Large Multimodal Models (LMMs). By introducing the MMECoT benchmark, it provides a detailed and systematic assessment across six diverse domains, offering novel metrics to evaluate reasoning quality, robustness, and efficiency. The in-depth analysis of state-of-the-art LMMs uncovers valuable insights, such as the role of reflection mechanisms in enhancing CoT quality and the potential inefficiencies that arise in self-correction phases. This work fills a gap in multimodal reasoning research, offering both a practical evaluation suite and a foundation for future advancements in the field.

**Weaknesses.**

1. For math-related problems in the benchmark, there are often multiple correct solutions, meaning that there are several valid paths. In such cases, it becomes difficult to fully assess the accuracy of the CoT process.

2.Some models tested in the paper, such as LLaVA-OV, Qwen2-VL, and InternVL2.5, do not generate CoT processes autonomously. The paper does not describe how the CoT outputs for these models were obtained. If these models are indeed capable, could other open-source models, like DeepSeekVL2 and GLM-4V, also be tested?

3. The selection process for Key Step Annotation is not provided in the paper, and it is also unclear whether a QA corresponds to a single or multiple Key Step Annotations.

**Questions For Authors:**

Please see weakness and suggestions above.

**Relation To Broader Scientific Literature:**

The key contributions of this paper build on prior research in Chain-of-Thought (CoT) reasoning for Large Language Models (LLMs) but extend the focus to Large Multimodal Models (LMMs). Previous work on CoT in LLMs (e.g., GPT-4) has shown improved reasoning capabilities. This paper expands that by evaluating how CoT affects LMMs in multiple domains, revealing novel insights.

**Theoretical Claims:**

This work does not include proofs for its theoretical claims. However, based on the results provided by the authors, the findings appear to be reasonable.

---

> ### Author Rebuttal · Authors · 2025-03-31
>
> We sincerely appreciate your valuable comments. We find them extremely helpful and will incorporate them in the final version. We address each comment in detail, hoping to address your concerns.
>
> > **Q1: Accuracy concern of questions of multiple correct solutions**
>
> **We believe this concern is addressed in our paper.** Our methodology explicitly accounts for questions with multiple correct solutions in both annotation and evaluation phases:
>
> 1. **Annotation Phase**: As detailed in Lines 203-206 of the main paper, annotators are explicitly instructed to provide all key steps for all possible solution paths.
> 2. **Evaluation Phase**: Our evaluation framework accommodates multiple solutions:
>    - For recall computation (Lines 245-246 and Equation 1), we compute recall scores for all possible solutions and select the maximum as the final recall score
>    - For precision, relevance rate, and reflection quality, we provide GPT-4o with key step annotations for all possible solutions in the evaluation prompt, enabling proper assessment of the CoT process considering all valid approaches
>
> > **Q2: How is CoT obtained in models that don't generate it autonomously?**
>
> Thanks for your advice. **We have incorporated the illustration of how to obtain the CoT output in Line 257 - 261 in the right columns of the main paper.** Specifically, we employ a CoT prompt to instruct the model to first perform step-by-step reasoning and finally give the answer. As illustrated in Line 355-358, the CoT prompt is:
>
> > Please generate a step-by-step answer, include all your intermediate reasoning process, and provide the final answer at the end.
>
> We empirically find that the models that don't generate CoT automatically like LLaVA-OV could give a detailed reasoning process before providing the final answer.
>
> GLM-4v cannot receive multiple images as input, so we only evaluate DeepSeek-vl2. The result is:
>
> Model|Pre.|Rec.|Eff.|Sta.|Rel.|Ref.
> -|-|-|-|-|-|-
> DeepSeek-VL2|81.2|43.0|-1.6|-5.1|93.3|100
>
> Please refer to the detailed table in Table 1 in https://anonymous.4open.science/r/reb-3538/README.md
>
> > **Q3: Selection process for key step annotation and the number of key step annotation for each QA**
>
> Thanks for your advice. We want to clarify the two points below:
>
> 1. **We have illustrated how to obtain the key steps in Line 187 - 205 in the main paper.** We detail the process below:
>
>    Key steps are defined as **necessary steps** for answering the question correcly, such as identifying critical values in math diagrams or reaching intermediate conclusions essential to determining the final answer.
>
>    To efficiently annotate key steps for all questions, we implemented a two-phase process. First, GPT-4o generated draft annotations using the questions, images, and final answers as inputs. Including the final answers significantly improved draft quality compared to using only questions and images. Second, human annotators reviewed these drafts, correcting any errors or developing key steps independently when GPT-4o failed to provide reasonable output.
>
>    All the key steps fall into two categories:
>
>    1. *Inference conclusions*: Necessary conclusions reached through logical inference steps (including the final answer)
>    2. *Image captions*: Identifications of critical visual information
>
>    We reduce all steps to their simplest form, preserving only core conclusions and relevant visual element descriptions. For problems with multiple solution paths, annotators are required to provide all possible methods.
>
> 2. **All QA corresponds to multiple key steps.**
>
>    1. **We provide a dataset visualisation in the bottom part of Fig. 2**. The key caption and key conclusion correspond to the inference conclusions and image captions of the key steps.  We will make this figure more clear in the final version.
>
>    2. **We also provide the statistics of the key step annotations in Table 1.** There are 837 reasoning questions with 3,865 key step annotations in total. On average, each question contains 4.6 key step annotations. This result can also be derived by the summation of the average inference conclusions and image captions listed in the table. We further look into each question and find out that all the questions have at least 3 key step annotations. We will make this more clear in the final version.
>
>
>    We will explicitly add this result in the table in the final version.

---

> > ### Comment · Reviewer_3D9g · 2025-04-04
> >
> > The authors have addressed my concerns. I will raise my score accordingly.

---

> > > ### Author Response · Authors · 2025-04-04
> > >
> > > Thank you very much for your reply! We commit to making the modifications mentioned in the rebuttal in the final version.

---

### Official Review · Reviewer_nZUW · 2025-03-13

**Overall Recommendation:** 3

**Summary:**

This paper introduces MME-CoT, a novel benchmark for evaluating chain-of-thought (CoT) reasoning capabilities in Large Multimodal Models (LMMs). The authors present a comprehensive evaluation framework that assesses three critical aspects of multimodal reasoning: quality, robustness, and efficiency. The insights derived from the experiment results are valuable and helpful to develop better CoT model.

## update after rebuttal
The authors's rebuttal has resolved most of my concerns. I hope the rebuttal contents can be added in the final version. I generally think this is a good paper and will left the AC to decide the acceptance

**Claims And Evidence:**

yes

**Essential References Not Discussed:**

No

**Experimental Designs Or Analyses:**

Yes. I checked section 4 for the details and the setting makes sense. Also Apendix B for prompt template

**Methods And Evaluation Criteria:**

Yes. This paper applies an LLM-as-judge evaluation method, and also provided LLM with the reference solution and background knowledge required to make the judgment. Therefore, although there might be concern on the hallucinations of LLM-as-judge, it's overall acceptable.

**Other Comments Or Suggestions:**

1. Are there any failure cases analysis in terms of the error type of the CoT, like calculation error, lack of background knowledge, etc? It's quite helpful for people to understand the behavior of the models on CoT generation.
2. If there are still resources, I would suggest the authors to conduct a human verification of the correctness of the GPT-4o as a judge. Although it's provided with the solution and each step along with the background knowledge, there is still a possibility that the GPT-4o can hallucinate. A ablation can alleviates concerns for this.

**Other Strengths And Weaknesses:**

### Strengths
Same as "Relation To Broader Scientific Literature"

### Weaknesses
1. Potential GPT-4o bias: The evaluation relies heavily on GPT-4o for judging various aspects of model outputs. This introduces potential biases, as GPT-4o itself may not be perfect at evaluating reasoning processes. This is a minor concern as the author already tried to give the users a reference solution and background knowledge.
2. There can be potential issues in the evaluation setting, where the comparison of CoT reasoning and direct reasoning can contain problems. Although for direct reasoning, the prompt is "Please directly provide the final answer without any other output", there is still a great change that the model ignore this instruction and keeps output the CoT, which is the behavior the model is trained on. Therefore, this can makes the comparison between CoT reasoning and direct reasoning meaningless. The authors should analyze and report the percentage of the response from direct reasoning that is actually direct. Otherwise, part of the conclusions like stability of the perception and reasoning tasks can be wrong conclusions.
3.

**Questions For Authors:**

1. What's the cost of evaluating a model since GPT-4o can be expensive.
2. The reflection quality of most models are 100, only Virgo-72B and QVQ-72B have pretty low scores. Are there any insights behind this result? Is that because we rarely observe reflection in other models?
3. For OCR tasks, how did GPT-4o judge the answer? The OCR response should only be regarded as correct when it's the same as the reference, otherwise 0. Will GPT-4o also give score if the response has pretty minor issue like missing a character or work, but the meaning is still the same?

**Relation To Broader Scientific Literature:**

1. This paper presents the first benchmark that evaluates the quality of CoT in terms of multiple aspects including precision and recall, also the robustness, etc. Previous benchmarks only focus on the outcome of the final answer and thus makes the paper's contribution significant.
2. This paper clearly distinguished the vision perception tasks from the reasoning tasks and performance analysis on them in different approaches. The insights that overthinking can do harm to the perception is a unique observation.

**Theoretical Claims:**

No. There is no theoritical claims in this paper.

---

> ### Author Rebuttal · Authors · 2025-03-31
>
> We sincerely appreciate your valuable comments. We find them extremely helpful and will incorporate them in the final version. We address each comment in detail, hoping to address your concerns.
>
> > **Q1: Potential GPT-4o bias in evaluation**
>
> Thank you for your valuable advice. We want to address your concern from two aspects:
>
> 1. **High Human-GPT-4o Alignment**
>
>    Our human alignment experiment shows strong correlation between GPT-4o evaluations and human judgment, confirming GPT-4o as a reliable tool for CoT evaluation. Specifically, we investigate two perspectives:
>
>    1. **Human Agreement Rate**: A binary (yes/no) human evaluation to assess agreement with the model's per-step judgments.
>    2. **Hallucination Detection**: We assess whether any reflection steps identified by GPT-4o are hallucinated or contain hallucinations.
>
>    We cover four key metrics: Recall, Precision, Relevance rate, and Reflection quality. We randomly sampled 54 predictions (9 questions from each subject) from QwenVL2-72B and QvQ-72B, totaling 216 predictions and 2,368 steps. The results are as follows:
>
>   Metric|Agreeement|Hallucination
>   -|-|-
>   Recall|98.5%|0%
>   Precision|94.1%|2.1%
>   Relevance rate|90.8%|0%
>   Reflection quality|86.1%|0%
>
> 2. **Best Available Automated Evaluation Method**
>
>    While acknowledging the inherent limitations, GPT-4o represents the current state-of-the-art approach for automatic CoT evaluation. GPT-4o has been widely adopted and validated for evaluation across various multimodal [1] and reasoning tasks [2]. Given our reference solutions and human alignment results, we believe GPT-4o is the most reliable option currently available for this complex evaluation task.
>
> > **Q2: Issues of true direct answers**
>
> Thanks for your valuable advice.  We report the actual direct answer (defined as less than 20 words) ratio in the table below:
>
> Model|Direct Ans Ratio
> -|-
> Mulberry|0.2600
> LLaVA-OV-7B|0.9712
> LLaVA-CoT|0.0000
> LLaVA-OV-72B|0.9704
> MiniCPM-V-2.6|0.9944
> InternVL2.5-8B|0.9920
> Qwen2-VL-7b|0.9776
> InternVL2.5-8B-MPO|0.9656
> InternVL2.5-78B-MPO|0.9992
> Qwen2-VL-72b|0.9776
> Virgo-72B|0.6728
> QVQ-72B|0.0008
> GPT4o|0.9648
>
> Four models demonstrate reluctance to provide direct answers (we mark these four models with \* in Table 2). Based on this, we recalculate stability and effectiveness using only questions that receive direct answers. The updated table is shown in Tab. 1 in https://anonymous.4open.science/r/reb-3538/README.md
>
> **We would like to emphasize that all conclusions in our current paper remain valid.** Our analysis and conclusion do not consider any results with \* to ensure validity. Thanks for pointing this out, and we will stress this further in the final version.
>
> > **Q3: CoT error analysis**
>
> Thanks for your advice. We provide the analysis of CoT error types and reflection error types.
>
> + CoT Error Types: Visual Perception, Visual Reasoning, Logical Reasoning, Calculation.
> + Reflection Error Types: Ineffective Reflection, Repetition, Incompleteness, Interference.
>
> Please refer to the example and error ratios in Fig. 3-6 in the link above.
>
> > **Q4: Human verification of GPT-4o correctness**
>
> Please refer to our response to Q1.
>
> > **Q5: The cost of evaluating GPT-4o**
>
> **The average cost of evaluating one model is 16 dollars**. We have also experimented with evaluating with GPT-4omini for all the tasks, but the human evaluation shows low agreement. With the recent development of LLM, the open-source models might be used for evaluation in the future.
>
> **We also plan to release a testmini set**, which composes 200 questions with all the subjects included. **Evaluation using testmini can reduce evaluation costs to around 3.5 dollars.**
>
> > **Q6: Most models score 100 in reflection quality**
>
> Yes, as illustrated in Line 381-382, **for models not generating reflection, we define their reflection quality to be 100** since the absence of reflection can be viewed as the most efficient approach.
>
> > **Q7: How does GPT-4o judge the answer for OCR tasks?**
>
> Thanks for your advice. We look into 20 predictions and their corresponding evaluation results. We observe that:
>
> 1. The tested models typically only OCR the content directly relevant to the question rather than performing comprehensive OCR on all text. The models tend to summarize other visual information briefly. This results in concise OCR content in predictions.
> 2. **We observe that GPT-4o demonstrates high accuracy in judging OCR content**, likely because the OCR content is usually highlighted in the key step annotations. Nevertheless, **we will still enhance our evaluation prompt to explicitly instruct GPT-4o to follow the OCR metric design you suggested**, ensuring that OCR responses are only considered correct when they exactly match the reference.
>
> [1] Tarsier: Recipes for training and evaluating large video description models
>
> [2] Mathverse: Does your multi-modal llm truly see the diagrams in visual math problems?

---

### Official Review · Reviewer_yz6o · 2025-03-14

**Overall Recommendation:** 3

**Summary:**

This paper introduces MME-CoT, a new benchmark for evaluating Chain-of-Thought (CoT) reasoning in Large Multimodal Models (LMMs). The work addresses a timely and important gap in the evaluation of multimodal reasoning. The authors have identified key limitations in existing benchmarks and propose a more comprehensive evaluation suite.

**Claims And Evidence:**

All the claims are problematic because the metrics themselves are not verified by human at all. Moreover, without any confidence interval, it is too early to draw any conclusion in a statistically significant manner.

**Essential References Not Discussed:**

As mentioned in Methods And Evaluation Criteria, references providing approaches to verify agreement with human shall be leveraged and cited.

**Experimental Designs Or Analyses:**

Please check details in Methods And Evaluation Criteria.

**Methods And Evaluation Criteria:**

The reviewer is really concerned about the validness of the current evaluation suite.

1. The evaluation relies heavily on GPT-4o(mini) for assessing various aspects of CoT (e.g., Recall, Precision, Relevance). While GPT-4o(mini) is a strong LLM, it's not perfect. Analysis of the agreement between LMM's judgments and human evaluations is necessary. For example, performing similar studies on human agreement in [a,b] could really make the work much more sound.

2. Confidence interval is necessary to draw meaningful conclusions to illuminate future research.

3. Current metrics have a rather small separation between models.

4. The current experimental setup is too simple, lack of sufficient variants to perform in-depth analysis. For example, to understand the importance of reflection/length of CoT, it is expected to at least try different prompts to encourage or discourage the model to include certain behavior in the prompts and perform more controlled experiments.

[a] MLLM-as-a-Judge: Assessing Multimodal LLM-as-a-Judge with Vision-Language Benchmark
[b] Personalized Video Comment Generation

[Update]: The reviewer appreciates the rebuttal and raised score accordingly based on the promised results. But given the current agreement value and the separation between models. The reviewer is still concerned on how this metric effectively reflect the actual rank between models and would like to call this out for ACs to consider.

**Other Comments Or Suggestions:**

NA

**Other Strengths And Weaknesses:**

Lack of more evaluation of other most capable models such as Gemini and Claude.

**Questions For Authors:**

Please check above for details.

**Relation To Broader Scientific Literature:**

N/A

**Theoretical Claims:**

N/A

---

> ### Author Rebuttal · Authors · 2025-03-31
>
> We sincerely appreciate your valuable comments. We find them extremely helpful and will incorporate them in the final version. We address each comment in detail, hoping to address your concerns.
>
> > **Q1: Need studies on human agreement**
>
> Thank you for your advice. As you suggested, we conduct additional human evaluations to verify the validity of the GPT-4o assessment from two perspectives [1, 2]:
>
> 1. **Human Agreement Rate**: A binary (yes/no) human evaluation to assess agreement with the model's per-step judgments.
> 2. **Hallucination Detection**: We assess whether any steps identified by GPT-4o are hallucinated or contain hallucinations.
>
> Our human agreement study covers four key metrics: Recall, Precision, Relevance rate, and Reflection quality. We randomly sample 54 predictions (9 questions from each subject) from QwenVL2-72B and QvQ-72B, totaling 216 predictions and 2,368 steps. The results are as follows:
>
> Metric|Agreeement|Hallucination
> -|-|-
> Recall|98.5%|0%
> Precision|94.1%|2.1%
> Relevance rate|90.8%|0%
> Reflection quality|86.1%|0%
>
> These results demonstrate a high correlation between GPT-4o evaluations and human judgment, indicating that GPT-4o is a reliable tool for CoT evaluation. This result also indicates that all of our analysis and conclusion are valid.
>
> We will incorporate references to the papers you suggested in our final version.
>
> > **Q2: Confidence interval is not provided**
>
> Thanks for your advice. Our default experiment setting uses temperature 0 for GPT-4o to ensure reproducible results. To address your concern, we set the temperature to 1 and evaluate 50 predictions for 5 times. The resulting 95% confidence interval widths are below:
>
> Pre.|Rec.|Eff.|Sta.|Rel.|Ref.
> -|-|-|-|-|-
> 0.018|0.026|0.004|0|0.049|0.069
>
> These narrow confidence intervals demonstrate the statistical reliability of our findings. The largest interval width is only 0.069 for Reliability, indicating that our results are stable and reproducible even with stochastic sampling.
>
> > **Q3:  Small separation between models of the current metric**
>
> We respectfully disagree with the reviewer's assessment for the following reasons:
>
> 1. **MME-CoT separation is comparable to other benchmarks:**
>
>    The difference is listed below:
>    Benchmark|Difference
>    -|-
>    F1-score (MME-CoT)|3.05
>    MMMU|0.83
>    MathVista|1.43
>    MathVerse|3.47
>
> 2. **The robustness and efficiency scores differ from traditional metrics, so the scale of the model's separation is also different.**
>
>    + The robustness score measures the performance difference within different prompts. This difference results from the intrinsic model attributes.
>    + The efficiency score specifically identifies models with excessively long CoT and reflection steps (e.g., QvQ and Virgo). As shown in Table 2, these two models score over 20 points lower in CoT efficiency compared to others, demonstrating significant separation.
>
> > **Q4: Experimental setup is too simple**
>
> We respectfully disagree that our experimental setup is too simple:
>
> 1. **Our experimental setup is sophisticated and comprehensive**
>
>    Compared with previous multimodal evaluation works [3, 4], our study explores multiple dimensions: different prompt strategies (CoT prompt vs. direct prompt), evaluation methods (CoT evaluation vs. direct evaluation), and diverse evaluation aspects (quality, robustness, and efficiency).
>
> 2. **Our focus is assessing natural CoT behavior, not improving it**
>
>    The primary objective of our paper is to evaluate how current LMMs perform reasoning when confronted with problems and to assess the quality of their reasoning processes. We specifically examine how models behave with standard prompting approaches rather than engineering prompts to encourage specific behaviors. The latter, while valuable, is beyond our current scope and for future research.
>
> > **Q5: Results of Claude and Gemini**
>
> Thanks for your advice. The results of Claude and Gemini are listed below:
>
> Model|Pre.|Rec.|Eff.|Sta.|Rel.|Ref.
> -|-|-|-|-|-|-
> Gemini-2.0-Flash|80.3|52.9|6.6|5.9|95.5|100
> Claude-3.5|77.2|48.2|9.9|11.0|91.0|100
>
> Please refer to the updated table in Table 1 in https://anonymous.4open.science/r/reb-3538/README.md
>
> [1] MLLM-as-a-Judge: Assessing Multimodal LLM-as-a-Judge with Vision-Language Benchmark
>
> [2] Personalized Video Comment Generation
>
> [3] Mathverse: Does your multi-modal llm truly see the diagrams in visual math problems?
>
> [4] Mmmu: A massive multi-discipline multimodal understanding and reasoning benchmark for expert agi

---

> > ### Comment · Reviewer_yz6o · 2025-04-04
> >
> > The reviewer appreciates the rebuttal and raised score accordingly based on the promised results. But given the current agreement value and the separation between models. The reviewer is still concerned on how this metric effectively reflect the actual rank between models and would like to call this out for ACs to consider.

---

> > > ### Author Response · Authors · 2025-04-05
> > >
> > > Thank you very much for your reply! We commit to including the experiments conducted and citing the related work in the rebuttal in the final version.

---

### Official Review · Reviewer_NFBU · 2025-03-14

**Overall Recommendation:** 3

**Summary:**

The paper introduces MME-CoT, a specialized benchmark evaluating the CoT reasoning performance of LMMs.

It is the first comprehensive study in LMM CoT evaluation: it spans six domains: math, science, OCR, logic, space-time, and general scenes; and it proposes a thorough evaluation suite incorporating novel metrics about reasoning quality, robustness, and efficiency.

In-depth analysis of state-of-the-art LMMs are shown, e.g., CoT performance on perception-heavy tasks, inefficiency of reflection mechanism.

**Claims And Evidence:**

Main claim: existing evaluation for LMM CoT is insufficiently systematic and thorough, while the proposed MME-CoT is a comprehensive and specialized benchmark for evaluating the CoT reasoning skills within LMMs.

The claim is clear and convincing. MME-CoT spans six fundamental domains and introduces comprehensive metrics. Also, diverse typical LMMs are evaluated on MME-CoT and analyzed.

**Essential References Not Discussed:**

None.

**Experimental Designs Or Analyses:**

I have checked experimental designs and analyses.

**Methods And Evaluation Criteria:**

The proposed metrics are sound for CoT evaluation.
The curated dataset spans six important domains and contains large-scale test data (1,130 questions with 3,865 key reasoning steps).
Experiments have been conducted for several typical LMMs.

**Questions:**

1. For CoT quality evaluation, it is difficult to understand L186-204 and identify the difference compared with existing metrics.

2. For CoT robustness evaluation, the paper proposes stability score based on perception tasks and efficacy score based on reasoning tasks. What about tasks involving both perception and reasoning? For example, counting red balls in a picture of balls with multiple colors, reasoning the relationship of two people from their activity, expression and scene. Such visual reasoning tasks are typical and important.

3. For CoT efficiency evaluation, a reflection should be "valid", either correctly pointing out the previous mistakes or verifying the previous conclusion with a new method. How to judge its validity? It needs comprehensive judgement and seems beyond capabilities of automatic evaluation.

**Other Comments Or Suggestions:**

1. Comparison with existing metrics can be shown in figures to better illustarte the novelty of MME-CoT benchmark.

2. More visualization and examples can be shown in the supplementary to support the analysis and conclusion.

**Other Strengths And Weaknesses:**

Strengths:
The paper is well-organized. The figures and tables are clear and easy to understand.

**Questions For Authors:**

My questions are mainly about metric design, which are listed in the "Methods And Evaluation Criteria" part.

**Relation To Broader Scientific Literature:**

The key contributions of the paper is related to LMM CoT evaluation.

**Theoretical Claims:**

No Theoretical Claims.

---

> ### Author Rebuttal · Authors · 2025-03-31
>
> We sincerely appreciate your valuable comments. We find them extremely helpful and will incorporate them in the final version. We address each comment in detail, hoping to address your concerns.
>
> > **Q1: Difficulty in understanding Sec 2.2 and difference with existing metrics**
>
> Thanks for your valuable advice. **We improve the clarity of Sec 2.2 below:**
>
>   For CoT evaluation, we provide *key steps* and *reference image captions* for all questions.
>
>   + *Key steps* are defined as **necessary steps** for answering the question correcly.
>
>     All the key steps fall into two categories:
>
>     1. *Inference conclusions*: Necessary conclusions reached through logical inference steps (including the final answer)
>     2. *Image captions*: Identifications of critical visual information
>
>     To efficiently annotate key steps for all questions, we implement a two-phase process. First, GPT-4o generates initial versions of key steps annotations (containing both inference conclusions and image captions) with the questions, images, and final answers as inputs. Including the final answers significantly improves the quality of the initial versions compared to using only questions and images. Second, human annotators review these initial versions, correcting any errors or developing key steps independently when GPT-4o failed to provide reasonable output.
>
>     We reduce all steps to the simplest form, preserving only core conclusions and relevant visual element description. For problems with multiple solution paths, annotators are required to provide all possible methods.
>
>   + *Reference image captions* are visual information not covered by the image caption in key steps.
>
>     These reference captions are mainly for the calculation of precision. We use the same method as key steps to obtain the annotation: GPT-4o first generates initial version and then annotators review and correct the errors.
>
>
> **Difference with existing metrics:**
>
> Few works evaluate the multimodal CoT process. MathVerse [1] represents one such effort. MME-CoT differs in several key aspects:
>
> 1. **MME-CoT evaluates the CoT quality in two ways: precision and recall**, corresponding to CoT faithfulness and informativeness. While [1] only instructs GPT-4 to judge the correctness of step, which could be viewed as only evaluating precision.
> 2. **MME-CoT contains ground truth key steps annotation**, enabling more reliable evaluation, especially for tasks beyond GPT-4o ability. [1] contain no such GT reference.
> 3. **MME-CoT considers both inference conclusions and image captions in CoT on all visual reasoning tasks**. [1] only targets on math tasks without identifying different types of CoT steps.
>
> > **Q2: Need to consider tasks involving both perception and reasoning**
>
> Sorry for the confusion caused. We'd like to clarify our task definitions:
>
> + The perception tasks are the tasks primarily test visual recognition abilities or require very minimal reasoning.
>
> + The reasoning tasks additionally require logical inference steps along with visual perception tasks. Therefore, **tasks requiring both perception and reasoning are exactly the reasoning tasks in MME-CoT**.
>
>   Examples in Fig. 2 (bottom section) of our paper all showcase this two requirements: first perceiving visual cues, and then reasoning based on the perception.
>
> + **For your examples**: Counting red balls belongs to perception task since it requires minimal reasoning. Determining relationship between people belongs to reasoning task. Similar reasoning task also occurs in MME-CoT, as shown in Fig. 1 in https://anonymous.4open.science/r/reb-3538/README.md
>
> > **Q3: The reflection evaluation seems to be beyond the model's capability**
>
> Thanks for your valuable advice. We identify GPT-4o as an well-qualified evaluator for assessing the reflection quality:
>
> 1. **GPT-4o shows competitive results of reflection quality evaluation in the human agreement experiments.**
>
>    We conduct additional human evaluations to verify the validity of GPT-4o assessment from two perspectives:
>
>    1. **Human Agreement**: A binary (yes/no) human evaluation to assess agreement with the model's per-step judgments.
>    2. **Hallucination Detection**: We assess whether any reflection steps identified by GPT-4o are hallucinated or contain hallucinations.
>
>    We randomly sample 54 predictions (9 questions from each subject) from QvQ-72B. The results are below:
>
> Agreeement|Hallucination
> -|-
> 86.1%|0%
>
> 2. **Additional instruction for reflection quality evaluation prompt.**
>
>    Acknowledging the challenges in identifying valid reflections, we incorporate specialized prompt design. We identify and list common reflection errors to better guide validity assessment (detailed in Lines 906-910).
>
> > **Q4&5: Comparison with existing metrics and more examples**
>
> We provide the comparison in Fig. 7 and more examples in Fig.8-18 in the link above.
>
> [1] Mathverse: Does your multi-modal llm truly see the diagrams in visual math problems?

---

### Decision · Program_Chairs · 2025-05-01

**Decision:**

Accept (poster)

**Comment:**

This paper presents MME-CoT, the first comprehensive benchmark designed to evaluate Chain-of-Thought (CoT) reasoning in Large Multimodal Models (LMMs) across six diverse domains using three novel metrics for reasoning quality, robustness, and efficiency. The study reveals that while reflection-based models like QVQ achieve high CoT quality, they often struggle with efficiency and may underperform on perception-heavy tasks due to overthinking prompted by CoT.


There was considerable discussion around this paper. The reviewers raised several concerns, including the heavy reliance on a single evaluation method (GPT-4o-mini), the absence of confidence intervals, and various aspects of the evaluation setup. The authors were highly active in the discussion and even questioned the expertise of the reviewers, something not uncommon in cases of disagreement. Overall, I align with the reviewers' consensus: this is an interesting paper with a potentially valuable and focused contribution. If there is space in the program, it would make sense to accept it.